# How Learning Rate Decay Wastes Your Best Data in Curriculum-Based LLM Pretraining

**Kairong Luo**[1][*]   **Zhenbo Sun**[1]   **Haodong Wen**[1]   **Xinyu Shi**[1]
**Jiarui Cui**[1]   **Chenyi Dang**[1]   **Kaifeng Lyu**[1][†]   **Wenguang Chen**[1,2][†]
[1]Tsinghua University    [2]Peng Cheng Laboratory

## Abstract

Due to the scarcity of high-quality data, large language models (LLMs) are often trained on mixtures of data with varying quality levels, even after sophisticated data curation. A natural approach to better leverage high-quality data is *curriculum-based pretraining*, where the model is trained on data sorted in ascending order of quality as determined by a quality metric. However, prior studies have reported limited improvements from such curriculum-based pretraining strategies. This work identifies a critical factor constraining these methods: the incompatibility between the ascending data quality order and the decaying learning rate (LR) schedule. We find that while curriculum-based training substantially outperforms random shuffling when using a constant LR, its advantage diminishes under standard LR decay schedules. Our experiments show this incompatibility can be mitigated by two simple strategies: (1) employing a more moderate LR decay schedule, where the final LR is only moderately smaller than the peak LR, and (2) replacing LR decay with model averaging, i.e., computing a weighted average of the final few checkpoints. By combining these strategies, we improve the average score on a suite of standard benchmarks by $1.64\%$ over random shuffling, without additional data refinement. Validated on 1.5B-parameter models trained over 30B tokens with various data-quality metrics, our findings call for a re-evaluation of curriculum-based LLM pretraining and underscore the potential of co-designing data curricula with optimization methods.

## 1 Introduction

Large language models (LLMs) are typically trained on massive text corpora collected from the Internet (Dubey et al., 2024; DeepSeek-AI et al., 2024; Yang et al., 2025; OpenAI, 2023), covering a wide range of sources and quality levels. High-quality data plays a crucial role in enhancing model capabilities, but it is usually limited in amount. To address this issue, current LLM pretraining pipelines employ sophisticated data curation procedures to filter out low-quality data and increase the proportion of high-quality data, including rule-based (or heuristic-based) filtering, quality scoring (model-based labeling), and score-based data selection (Su et al., 2025; Li et al., 2024; Penedo et al., 2025; 2023; Weber et al., 2024). Despite these advances, relatively little attention has been given to developing *training strategies* that more effectively utilize the high-quality data during training, rather than only during data curation.

A natural idea to improve the utilization of high-quality data is to use *curriculum learning*[1]. This is motivated by the catastrophic forgetting problem (McCloskey & Cohen, 1989), which refers to the phenomenon that a model may forget the knowledge it has learned before when it is exposed to new data (Dai et al., 2025; Liao et al., 2025). In contrast to random shuffling, this curriculum-based approach aims to optimize knowledge acquisition by exposing the model to high-quality data in the latter stages of training.

---

[*]luokr24@mails.tsinghua.edu.cn
[†]Corresponding authors.
[1]Traditionally, curriculum learning refers to training on progressively harder examples. Here, following prior work (Wettig et al., 2024), we generalize the term to denote data-ordering strategies such as progressing from low-quality to high-quality data.

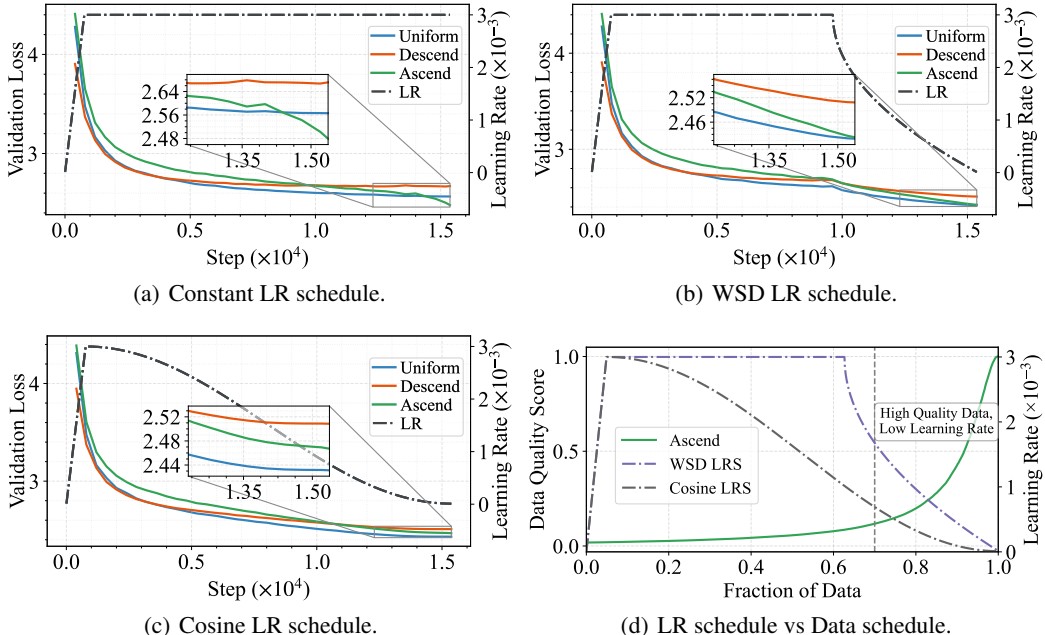

Figure 1: Data curriculum strategies are less effective when combined with learning rate (LR) schedules that decay to a low scale near the end. **(a-c)** Experiments on a 1.5B parameter model trained on 30B tokens compare various data curricula (Uniform, Ascending-Order, and Descending-Order by DCLM score (Li et al., 2024)) under constant, Warmup-Stable-Decay (WSD) (Hu et al., 2024; Hägele et al., 2024), and cosine schedules. While curricula improve validation loss over a uniform baseline with a constant LR, this advantage is significantly reduced during a low-LR phase following LR decay. **(d)** In the data curriculum, high-quality data is placed in the latter phase, which coincides with the LR decaying to a relatively low scale.

One successful curriculum learning strategy is multi-stage pretraining: first training on a data mixture dominated by massive web data, then in the second stage, referred to as *mid-training* (OLMo et al., 2025; Abdin et al., 2024b), shifting the data mixture to one that mainly consists of high-quality data. This strategy has been adopted by many recent LLMs, including OLMo 2 (OLMo et al., 2025), Phi-4 (Abdin et al., 2024a), and LongCat-Flash (Team et al., 2025). This two-phase design is most common, and it is also promising to extend with more stages (Yiwen et al., 2025; Allal et al., 2025) or follow with long-context extension (Yang et al., 2025).

Another line of work explores curriculum learning at the *instance level*, where data samples are sorted according to quality scores and presented to the model sequentially (Wettig et al., 2024; Dai et al., 2025; Zhang et al., 2025; Kim & Lee, 2024). We refer to this as the *data curriculum*[2]. However, these studies mainly investigate different quality metrics and find that simple end-to-end sorting yields limited benefits. Consequently, several works propose alternative strategies such as *folding curriculum* (Detailed in Section 2), which reorders samples within consecutive phases in an interleaved manner (Dai et al., 2025; Zhang et al., 2025). Despite showing promise, we find that this interleaved approach is fragile: its advantage does not extend to our larger-scale experiments with the DCLM fastText score (Li et al., 2024), a widely used scoring metric (see Section 2).

This raises a central question: *Why do instance-level curriculum learning strategies often yield limited benefits?* This is not simply due to unreliable quality scores: metrics like the QuRating score (proposed and used by Wettig et al. (2024)), the PDS score (proposed by Gu et al. (2025b) and discussed by Dai et al. (2025)), and the DCLM score (Li et al., 2024) are already informative enough to improve training efficiency by guiding high-quality data selection.

**Our Contributions.** In this paper, we identify a key, yet previously overlooked factor: *the incompatibility between the ascending order of data quality and the decaying schedule of learning rate*. As illustrated in Figure 1, if we train an LLM with a constant LR, using a data curriculum that sorts data in ascending order of quality can indeed outperform the baseline that trains the model on data in a uniform order. However, when we switch to a more standard LR decay schedule, such as cosine

---

[2]We use *data schedule* as a general term for any strategy that specifies the order of training data. Unless stated otherwise, *data curriculum* denotes a schedule that sorts data samples in ascending or descending order (reverse curriculum) with respect to a particular quality metric.

or Warmup-Stable-Decay (WSD) (Loshchilov & Hutter, 2017; Hu et al., 2024) (a schedule with warmup, plateau, and decay phases, see Figure 1(b)), the benefit of the data curriculum diminishes. Moreover, we observe that as the LR decay becomes more aggressive (e.g., having a longer decay phase or a lower ending LR), the benefit of the data curriculum diminishes more.

To resolve the incompatibility between data curriculum and LR decay, firstly, we discuss a straightforward remedy: adopting a moderate LR decay schedule in curriculum learning. In Section 3.1, we show that tuning the ending LR in WSD schedules trades off the benefits of data curriculum and loss convergence, and setting it to a moderate value (approximately decaying to $1/3$ of peak LR in our setting) can make use of high-quality data and outperform uniform data ordering.

Further, we propose a strategy that largely resolves the incompatibility between the data curriculum and the LR schedule by replacing LR decay with *weight averaging* (Li et al., 2025c; Izmailov et al., 2018; Tian et al., 2025). Weight averaging computes a weighted average of recent checkpoints as the final model. It stabilizes model parameters and reduces noise in the training process, similar to the effect of LR decay. However, it can achieve this without diminishing the magnitude of updates. Therefore, we adopt a constant LR throughout training and pair weight averaging with the data curriculum. This combination allows the model to maintain a high learning rate and fully exploit the high-quality data introduced later in the curriculum. We call this approach *Curriculum Model Averaging (CMA)*. Notably, CMA also extends emerging practices of introducing weight averaging into LLM training (Tian et al., 2025; Li et al., 2025c; Sanyal et al., 2024), showing that combining weight averaging with curriculum learning yields greater benefits than applying weight averaging to standard uniform-ordering pretraining. This combination is particularly effective under multi-phase pretraining, where high-quality data is introduced in the mid-training phase. In this setting, our approach achieves an average improvement of 1.2% in accuracy—and over 2% on core benchmarks (defined in Section 2)—solely by reordering data samples (Table 2).

Building on these explorations, we further demonstrate that combining moderate LR decay, curriculum learning, and weight averaging can produce synergistic advantages and reveal a previously overlooked high-performing pretraining regime, improving standard benchmarks by 1.64% in average accuracy over random shuffling and standard decay. This finding underlines that hyperparameter settings optimized for uniform data ordering are not necessarily optimal for curriculum-based training. Prior work has naturally evaluated curriculum-based training using regimes originally optimized for uniform data ordering. However, this regime is suboptimal for curriculum-based training and, as previously reported, leads to only marginal benefits (Wettig et al., 2024; Kim & Lee, 2024). In contrast, we identify a previously underexplored and more effective regime in the design space of LLM pretraining by co-designing these components, paving the way for future exploration.

We validate our hypotheses and proposed strategies through experiments at a scale sufficient to support our conclusions, training a 1.5 billion-parameter model on 30 billion tokens. Our findings demonstrate robustness across different quality metrics, LR schedules, and data mixtures, highlighting that co-designing data curricula with training dynamics is a powerful, data-aware strategy for improving LLM pretraining efficiency.

## 2 LEARNING RATE DECAY COUNTERACTS DATA CURRICULUM

In this section, we analyze the critical yet often overlooked interaction between the learning rate (LR) schedule and the data schedule. We first explain how the learning rate acts as an implicit importance weight for each data sample. We then present empirical results to demonstrate three key points: (1) a data curriculum can yield significant benefits over a uniform data order under a constant LR schedule; (2) these benefits diminish when a conventional decaying LR schedule is applied, particularly during the final, high-quality data regime; and (3) while adjustments to the data schedule can mitigate this issue, the underlying conflict persists.

**Analysis of Coupling between Learning Rate and Data Schedules.** A key insight is that the learning rate schedule acts as an implicit importance weight for each training sample. The parameter update at training step $t$ is $\boldsymbol{\theta}_{t+1} = \boldsymbol{\theta}_t - \eta_t \boldsymbol{g}_t$, where $\eta_t$ is the learning rate. The gradient $\boldsymbol{g}_t$ can be decomposed into a signal component, $\mathbb{E}[\boldsymbol{g}_t]$, which points in the direction of steady improvement, and a noise component, $\epsilon_t$. A decaying learning rate $\eta_t$ serves a dual purpose: it reduces the noise $\epsilon_t$ to stabilize training, but it also shrinks the update step taken in the signal direction $\mathbb{E}[\boldsymbol{g}_t]$. While modern optimizers like Adam (Kingma & Ba, 2015) use more complex update rules, the learning rate remains a dominant factor in the update magnitude. This dual role of $\eta_t$ creates a fundamental

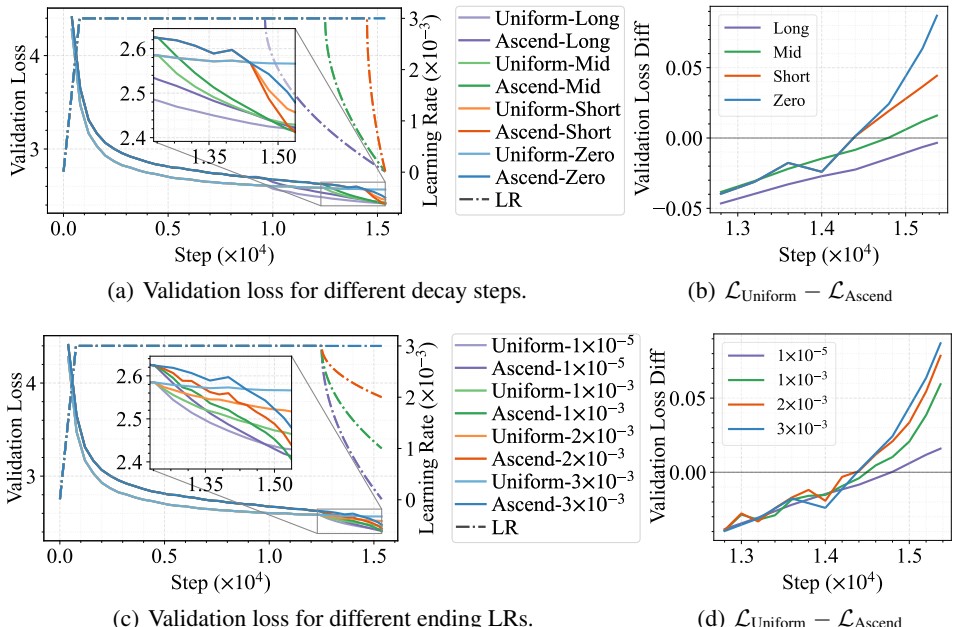

Figure 2: When varying the decay steps across 37%, 18%, 6% and 0% of training (*Long*, *Mid*, *Short*, *Zero*, respectively) and ending LRs ($1 \times 10^{-5}, 1 \times 10^{-3}, 2 \times 10^{-3}, 3 \times 10^{-3}$), the benefit of data curriculum diminishes with more aggressive LR decay. For each LR decay, we train 1.5B-parameter models with uniform and ascending ordering of data based on DCLM scores, and measure the difference in validation loss. As shown in (b) and (d), this difference becomes smaller with more decay steps or smaller ending LRs.

conflict in quality-based curricula. High-quality samples are intentionally processed at the end of training, but this is precisely when conventional LR schedules reduce $\eta_t$ to its minimum. Consequently, the decaying learning rate diminishes the influence of the most valuable data, counteracting the intended benefit of the curriculum.

**Experimental Settings.** Our experiments are grounded in the DataComps-LM (DCLM) framework (Li et al., 2024) at the `1B-1x` scale, ensuring our findings are validated at a substantial scale. We adopt the Qwen2.5-1.5B model architecture (Yang et al., 2024a) and train models on a 30B token subset of the DCLM-Baseline dataset. For the data curriculum, we use DCLM's fasttext scores as our quality metric. We set the peak LR to $3 \times 10^{-3}$ and the ending LR to $1 \times 10^{-5}$, aligning with optimal settings found in prior work for uniform data schedules (Li et al., 2024; Luo et al., 2025; Li et al., 2025b). We evaluate performance on a high-quality subset of the DCLM-Baseline dataset held out for validation. We also report downstream task scores by the OLMES framework (Gu et al., 2025a). Among the standard benchmark suite, we choose MMLU (Hendrycks et al., 2021), ARC (Clark et al., 2018), and CSQA (Talmor et al., 2019) as *Core* benchmarks, as recent work suggests they have a higher signal-to-noise ratio to distinguish model performance (Heineman et al., 2025). Further experimental details are provided in Section C.1.

**A Data Curriculum is Highly Effective with a Constant Learning Rate.** To isolate the effect of the data schedule from the LR schedule, we first conducted experiments using a constant learning rate of $3 \times 10^{-3}$. We compared three data schedules: a uniform random baseline, an ascending-order curriculum, and a reverse (descending-order) curriculum, both curricula sorted by DCLM quality scores. As shown in Figure 1(a), the ascending-order curriculum significantly outperforms the uniform baseline, achieving a much lower validation loss and faster convergence. In contrast, the reverse curriculum's validation loss trends upward, likely because the data distribution shifts progressively away from the high-quality validation set. These results clearly demonstrate that a quality-based curriculum is effective when its impact is not confounded by a decaying learning rate. Similar trends were observed using PreSelect scores (SHUM et al., 2025) (see Appendix Figure 8(a)).

**The Curriculum's Advantage Diminishes with a Decaying LR Schedule.** In sharp contrast to the constant LR experiments, the advantage of the data curriculum largely disappears when we employ a WSD schedule (Figure 1(b)). The performance degradation is even more pronounced with a cosine schedule, which decays throughout its entire range (Figure 1(c)). To further probe this relationship, we varied the aggressiveness of the LR decay by adjusting two WSD parameters: the

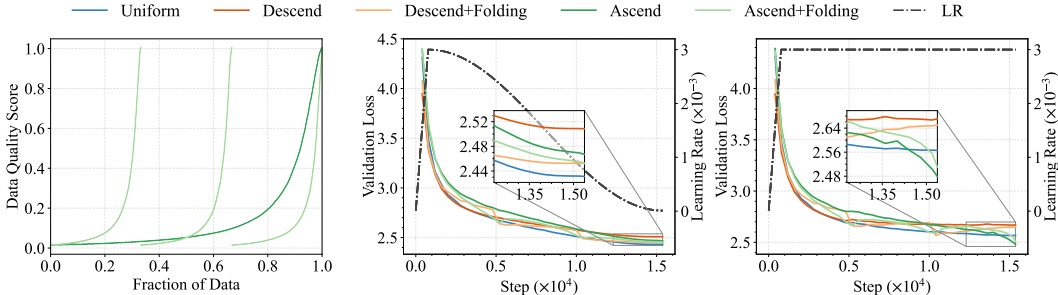

Figure 3: A stage-wise "data folding" curriculum mitigates the negative interaction observed between data ordering and learning rate (LR) decay (detailed in Section 2), but data folding can not match end-to-end sorting under a constant learning rate. **Left:** We compare simple ascending curricula (Ascend), sorted by DCLM score, against their "folding" counterparts (Ascend+Folding). The folding method involves partitioning the data into stages (three in our implementation) and performing the sort within each stage. The Descend(+Folding) curriculum is designed in reverse order. **Middle:** Under a standard cosine LR schedule, folding strategies reduce validation loss compared to simple sorting but are outperformed by a uniform data baseline. **Right:** Conversely, with a constant LR schedule where decay does not weaken the utility of high-quality data, the advantage of folding vanishes, and a simple ascending-order curriculum becomes the most effective strategy.

number of decay steps and the final learning rate. The results in Figure 2 show a clear trend: as the decay phase becomes longer or more aggressive (i.e., a smaller ending LR), the performance benefit of the data curriculum over the uniform baseline shrinks, eventually becoming negligible. This confirms the previously overlooked effect of LR decay on the data curriculum, where LR decay undermines the contribution of high-quality data.

**Exploring an Alternative Curriculum: Data Folding.** We also investigated whether a different curriculum design could mitigate the coupling effect. We tested a *folding* curriculum, inspired by prior work (Dai et al., 2025; Zhang et al., 2025), where the dataset is split into several chunks and each chunk is sorted internally. This stage-wise design distributes high-quality data more evenly throughout training than the standard data curriculum. As shown in Figure 3, under a cosine schedule, the ascending with folding strategy performed better than a simple end-to-end ascending sort but still underperformed the uniform baseline. Conversely, under a constant LR schedule, the simple ascending-order curriculum proved to be the most effective strategy, outperforming the folding curriculum. These results support our hypothesis: when the LR decays, folding offers a slight benefit over simple sorting by processing some high-quality data earlier, but under a constant LR, where this is not a concern, a simple end-to-end curriculum remains superior. Refer to Section E.1 for a more detailed discussion on data folding experiments.

## 3 UNLOCKING DATA CURRICULA POTENTIAL VIA MODERATE LR DECAY AND MODEL AVERAGING

To resolve the interaction between data curricula and learning rate (LR) schedules, we first utilize a more moderate LR decay in place of a standard aggressive decay, which we find mitigates the issue. We then turn to a more principled approach: *model averaging* (Izmailov et al., 2018; Li et al., 2025c; Tian et al., 2025). We investigate replacing LR decay entirely with model averaging, which allows high-quality data to be processed with a constant learning rate. While model averaging alone may not match the performance of LR decay with uniform data, we find that combining a data curriculum with model averaging produces comparable or even superior results to a standard decaying LR schedule, particularly in a mid-training setting. This reveals a synergistic relationship between data curricula and model averaging. Furthermore, we find that a combination of model averaging and moderate LR decay can yield even stronger and stable results for curriculum-based pretraining. Our results highlight a previously unexplored regime for improving LLM pretraining and reveal the potential of co-designing LR schedules, data curricula, and model averaging strategies.

### 3.1 MITIGATING NEGATIVE INTERACTION WITH MODERATE LEARNING RATE DECAY

A straightforward way to mitigate the negative impact of LR decay on data curricula is to use a moderate LR decay instead of a standard aggressive one. As shown in Figure 2, for uniform data, the validation loss decreases with more aggressive LR decay, like increasing decay steps and lower ending LRs in our experimental setting. In comparison, the benefits of data curricula increase with

Table 1: Curriculum Model Average (CMA) exhibits advantages over standard LR decay schedule pretraining, much better than the widely used *Cosine+Uniform* setting. Our proposed methods are highlighted in gray. **WA**: Weight Averaging technique (Section B). **Order**: Data ordering. **LRS**: Learning Rate Schedule (WSD: Warmup-Stable-Decay, Cos: Cosine, Const: Constant). **Core**: Average score on the first four, high signal-to-noise tasks according to prior work (Heineman et al., 2025) (MMLU, ARC-c, ARC-e, CSQA). Both the Core and Avg. scores are annotated with a subscript indicating the performance change relative to the baseline (*WSD + Uniform*). Performance changes are color-coded: **bold green** ($\geq 0.5$ improvement), light green ($> 0$ improvement), and red (decrease).

| WA | Order | LRS | MMLU | ARC-c | ARC-e | CSQA | Core | OBQA | PIQA | SIQA | Wino. | Avg. |
|---|---|---|---|---|---|---|---|---|---|---|---|---|
| ✗ | Uniform | Cos | 30.49 | 38.13 | 59.47 | 49.14 | $44.31_{-1.90}$ | 42.20 | 71.87 | 45.19 | 56.51 | $49.13_{-1.43}$ |
| ✗ | Ascend | Cos | 30.80 | 39.80 | 59.12 | 51.27 | $45.25_{-0.96}$ | 42.60 | 71.55 | 45.65 | 57.06 | $49.73_{-0.83}$ |
| ✗ | Uniform | WSD | 30.77 | 42.14 | 61.05 | 50.86 | 46.21 | 45.20 | 72.42 | 45.75 | 56.27 | 50.56 |
| ✗ | Ascend | WSD | 31.58 | 38.80 | 61.05 | 50.37 | $45.45_{-0.76}$ | 45.80 | 71.82 | 46.01 | 57.30 | $50.34_{-0.22}$ |
| WMA | Uniform | Const | 30.87 | 37.12 | 58.95 | 53.24 | $45.04_{-1.17}$ | 43.40 | 71.76 | 46.26 | 57.38 | $49.87_{-0.69}$ |
| SMA | Uniform | Const | 31.22 | 36.12 | 59.82 | 53.97 | $45.28_{-0.93}$ | 43.40 | 71.98 | 46.42 | 57.85 | $50.10_{-0.46}$ |
| EMA | Uniform | Const | 31.39 | 36.45 | 59.82 | 53.48 | $45.29_{-0.92}$ | 42.40 | 72.14 | 46.32 | 57.54 | $49.94_{-0.62}$ |
| WMA | Ascend | Const | 31.67 | 39.80 | 61.40 | 53.07 | $46.49_{+0.28}$ | 45.00 | 71.93 | 45.45 | 57.14 | $50.68_{+0.12}$ |
| SMA | Ascend | Const | 32.28 | 40.80 | 62.11 | 52.91 | $47.02_{+0.81}$ | 44.80 | 71.60 | 45.80 | 57.22 | $50.94_{+0.38}$ |
| EMA | Ascend | Const | 32.17 | 40.80 | 61.75 | 53.07 | $46.95_{+0.74}$ | 44.80 | 71.55 | 45.85 | 57.62 | $50.95_{+0.39}$ |

a more moderate LR decay, like fewer decay steps or higher ending LRs. The different preferences of LR decay suggest that the optimal ending LR or number of decay steps can differ for curriculum-based training from uniform data training, also indicated by the validation loss curves in Figure 2. Since the optimal ending LR is typically close to zero for uniform data ordering, but more decay steps are not always better (Li et al., 2025b; 2024; Hu et al., 2024), it is more convenient to ablate on the ending LRs to see whether the optimal LR schedule changes for curricula.

To investigate this, we run experiments on ending LRs in a fine-grained manner and report the training results for both uniform data ordering and a data curriculum. As shown in Figure 5(a), we find that the data curriculum (Ascend+WSD) may only achieve a marginal improvement or even fail to match the performance of uniform ordering (Uniform+WSD) when the ending LR is close to zero, like at the scale of $10^{-5}$. However, as the ending LR increases, the performance of curriculum training improves, but may degrade when the ending LR approaches the peak LR. Note that the performance of the data curriculum can decline while its relative benefit over uniform ordering can still increase as the uniform training performance degrades more sharply when ending LR increases. The optimal ending LR of the data curriculum is around $1 \times 10^{-3}$, much higher than that for uniform data ordering, and the tuned data curriculum training outperforms the optimal uniform data training results. These experiments validate that using a more moderate LR decay—for instance, adjusting the ending LR to approximately $1/3$ of the peak LR in our setting—can effectively mitigate the negative interaction and unlock the benefits of a data curriculum.

## 3.2 MODEL AVERAGE CAN HELP DATA CURRICULUM

**CMA: Replacing Learning Rate Decay with Model Averaging.** Adjusting the ending LR serves as a trade-off between the benefits of a data curriculum and LR decay. To fully utilize the benefits of data curricula, we propose to decouple the data schedule from the side effects of LR annealing by replacing LR decay entirely with model averaging. In this approach, we replace the decaying LR schedule with a constant LR and apply model averaging to the final checkpoints of the training process. We call this strategy **C**urriculum **M**odel **A**veraging (CMA), detailed in Algorithm 1. In our default setting, we use Exponential Moving Average (EMA) with $\alpha = 0.2$ over the last six checkpoints. The types of weight averaging considered include Simple Moving Average (SMA), EMA, and Weighted Moving Average (WMA), as introduced in Section B. For comparison, we use standard LR pretraining schedules, including cosine and WSD (introduced in Section B). The strongest baseline for our evaluation, denoted *WSD + Uniform*, is the best-performing combination of a standard LR schedule and data order. Downstream task performances are reported in Table 1, and practical implementation details are reported in Section C.2.

**The Synergy of Data Curriculum and Model Averaging.** The results in Table 1 lead to several key observations. First, the combination of a data curriculum and model averaging (e.g., *EMA + Ascend*) outperforms models trained with a standard LR decay schedule, including *WSD + Uniform* and *WSD + Ascend*. It also consistently outperforms model averaging applied to a uniform data order (e.g., *EMA + Uniform*). Second, this synergy is crucial, as other combinations yield limited improvements. For instance, model averaging with a uniform data order (*EMA + Uniform*) under a constant learning rate does not fully match a standard WSD schedule. Furthermore,

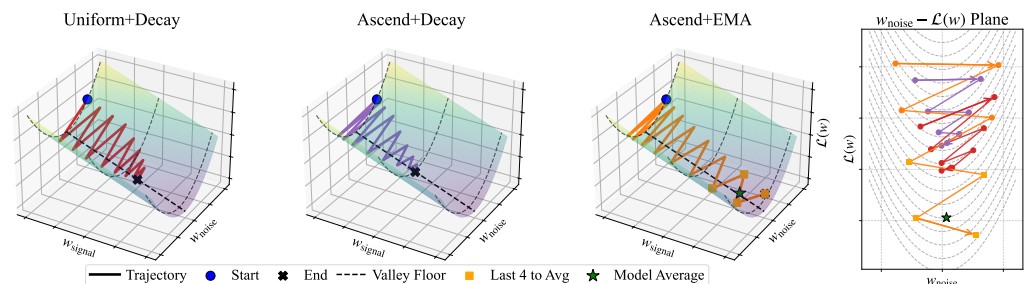

Figure 4: Visualization of our intuition about the interplay between data ordering and LR schedules. We assume the gradient update can be decomposed as a signal direction and a noise direction. High-quality data can offer a less noisy direction and a more stable signal direction, while low-quality data can induce a more noisy update. *Uniform+Decay*, *Ascend+Decay* and *Ascend+EMA* represent different training strategies. *Ascend+EMA* can make the best use of the high-quality data in the curriculum. The right-hand figure shows the projection of the trajectories of the last 8 steps for these cases onto the $w_{noise}$-$\mathcal{L}(w)$ plane.

combining a standard LR decay with a data curriculum (*WSD + Ascend*) provides only marginal gains and can even degrade performance, confirming the negative interaction we identified between schedules. These results highlight the necessity of combining both a data curriculum and weight averaging, a strategy largely overlooked by prior work that has focused on either weight averaging (Tian et al., 2025; Yang et al., 2024b) or curriculum design (Dai et al., 2025) in isolation. Third, aligning the checkpoint weights with the data schedule is beneficial: EMA and SMA, which assign non-decreasing weights to later (and higher-quality) checkpoints, outperform WMA under a data curriculum. The differences between these moving average strategies are detailed in Section B.

**Motivation: Synergy from Decoupling Schedules.** We intuitively interpret the synergy between a data curriculum and model averaging from a loss landscape perspective, emphasizing the interplay between two key factors: the learning rate and data quality. We present a visualization of this concept in Figure 4. The learning rate controls the size of the update steps, while data quality influences the signal-to-noise ratio of the gradients. *Uniform+Decay* progresses with a relatively consistent level of noise compared to the signal from the training data, and the final decay reduces the noise but also limits the update rate along the signal direction; *Ascend+Decay* starts with high noise, but the step-size decays too fast, so it does not utilize the good signal from high-quality data to move faster near the end; When it comes to the model averaging strategy, the training over uniform data may not reduce noise as effectively as a near-zero learning rate, which could explain its slightly lower performance reported in Table 1. However, when using a curriculum strategy, labeled as *Ascend+EMA*, although the training starts with higher noise, the high-quality data introduced late in training provides a clearer and more reliable gradient. This strategy maintains the update magnitude in the high-quality regime, allowing it to take advantage of the good signal near the end along the signal direction. Using model averaging can also reduce noise along the noise direction, probably achieving a better balance between progress and stability compared to aggressive learning rate decay. We also provide a simplified theoretical model in Section 4 and discuss our perspective in the context of prior work (Wen et al., 2025) in Section E.1.

### 3.3 RESULTS ON MID-TRAINING WITH MIXED QUALITY DATA

**CMA Benefits are More Pronounced in Mid-Training.** Mid-training is an emerging practice in LLM pretraining where a large corpus of average-quality data is supplemented by a smaller, high-quality dataset in a later training stage (Yang et al., 2025; OLMo et al., 2025; Hu et al., 2024). We conduct mid-training experiments, with settings detailed in Section C.1. As shown in Table 2, CMA exhibits a larger benefit in this practical setting compared to experiments on uniformly high-quality data. The CMA results (e.g., *EMA + A-T*) show a significant advantage over the WSD schedule baseline (*WSD + U,U*), improving the average accuracy by 1.20% and achieving over a 2.0% improvement on average across the core suite of benchmarks. This margin is notable given that no additional complex data filtering is applied. A possible explanation for the more prominent improvement is that, when high-quality data is sparse, each sample provides a relatively more valuable signal for parameter updates, amplifying the benefit of CMA.

**A Practical and Simplified Strategy also Performs Well.** In practice, sorting an entire data corpus globally may not be feasible. As an alternative, we tested a strategy where data is sorted in ascending order within each training phase separately (*A,A* in Table 2). The results show that the benefits of our approach over standard LR decay largely persist. However, applying a curriculum

Table 2: The benefit of CMA becomes more prominent in the mid-training setting. Our proposed methods are highlighted in gray. **WA**: Weight Averaging technique (Section B). **Order**: Data ordering in two phases (U: Uniform, A: Ascend). A-T (All-Together) sorts data samples in both phases as a whole. **LRS**: Learning Rate Schedule (WSD: Warmup-Stable-Decay schedule, Const: Constant LR). **Core**: Average score on the first four, high signal-to-noise tasks (MMLU, ARC-c, ARC-e, CSQA). Both the Core and Avg. scores are annotated with a subscript indicating the performance change relative to the baseline (*WSD + U,U*). Performance changes are color-coded: **bold green** ($\geq 0.5$ improvement), light green ($> 0$ improvement), and red (decrease).

| WA | Order | LRS | MMLU | ARC-c | ARC-e | CSQA | Core | OBQA | PIQA | SIQA | Wino. | Avg. |
|---|---|---|---|---|---|---|---|---|---|---|---|---|
| ✗ | U,U | WSD | 29.23 | 33.78 | 53.86 | 49.55 | 41.61 | 40.40 | 71.87 | 44.78 | 56.43 | 47.49 |
| ✗ | U,A | WSD | 29.44 | 34.45 | 52.63 | 50.12 | $41.66_{+0.05}$ | 41.00 | 71.76 | 44.42 | 56.75 | $47.57_{+0.08}$ |
| ✗ | A,A | WSD | 30.22 | 33.11 | 56.84 | 47.34 | $41.88_{+0.27}$ | 39.40 | 71.55 | 44.78 | 56.67 | $47.49_{0.00}$ |
| ✗ | A-T | WSD | 29.93 | 37.12 | 54.39 | 49.47 | $42.73_{+1.12}$ | 39.00 | 72.20 | 45.14 | 56.83 | $48.01_{+0.52}$ |
| EMA | U,U | Const | 29.84 | 32.78 | 52.28 | 51.52 | $41.60_{-0.01}$ | 42.00 | 71.60 | 44.68 | 56.99 | $47.71_{+0.22}$ |
| EMA | U,A | Const | 29.75 | 35.12 | 51.75 | 48.57 | $41.30_{-0.31}$ | 42.20 | 70.51 | 44.83 | 56.91 | $47.45_{-0.04}$ |
| EMA | A,A | Const | 30.31 | 36.45 | 57.54 | 50.12 | $43.61_{+2.00}$ | 41.40 | 72.14 | 45.09 | 56.43 | $48.69_{+1.20}$ |
| EMA | A-T | Const | 30.81 | 36.29 | 57.89 | 50.29 | $43.82_{+2.21}$ | 44.50 | 70.62 | 44.68 | 54.46 | $48.69_{+1.20}$ |
| SMA | A-T | Const | 30.65 | 36.79 | 57.37 | 50.78 | $43.90_{+2.29}$ | 43.60 | 70.89 | 44.73 | 54.74 | $48.69_{+1.20}$ |

only to the final, high-quality data phase (*EMA + U,A*) is not sufficient for optimal results. The superior performance of the *A,A* schedule over *U,A* suggests that applying a curriculum to the initial, lower-quality data phase is also beneficial. One possible explanation is that reordering data in the lower-quality phase can exploit the model's forgetting mechanism to mitigate the adverse effects of toxic samples that pass through the cleaning pipeline by chance. These results further confirm the synergy between model averaging and a data curriculum.

## 3.4 OVERLOOKED BENEFIT: CO-DESIGN OF DATA CURRICULUM, LR SCHEDULE, AND WEIGHT AVERAGE

**CDMA: Combining Moderate LR Decay with Model Averaging.** We have identified the benefits of the moderate LR decay and weight averaging in curriculum-based pretraining. A natural question is whether combining a moderate LR decay with model averaging under a data curriculum can yield further improvements. We conducted a series of experiments varying the ending learning rate in WSD schedules, from $1 \times 10^{-5}$ to $3 \times 10^{-3}$, and then applied EMA to the final checkpoints. As shown in Figure 5, this combination achieves stable and optimal results with a moderate LR decay (i.e., a higher ending LR than in standard practice). While a data curriculum with moderate LR decay alone also achieves near-optimal results, it does not fully match the combined strategy and may need a more careful tuning of ending LRs. These two curriculum-based strategies both outperform their uniform-ordering training counterparts when the LR decay is properly tuned for both curriculum-based and uniform-ordering. Furthermore, as shown in Figure 5(b) of the mid-training setting, the best combination can improve by 1.68% in the average benchmark accuracy over the baseline (Uniform+WSD with ending LR $1 \times 10^{-5}$, corresponding to the left endpoint of the blue dash line), a prominent improvement without any additional data refinement. This motivates the following guideline for curriculum-based pretraining: *use a moderate LR decay and adopt model averaging*. To distinguish this from CMA, we call this strategy **C**urriculum with LR **D**ecay **M**odel **A**veraging (**CDMA**). Specifically, curriculum-based LLM pretraining should use a more moderate LR decay than the optimal setting for uniform data training; their optimal regimes can differ substantially (around $1 \times 10^{-5}$ for uniform training versus roughly $1 \times 10^{-3}$ for curriculum-based training in our setting). A weight averaging strategy can further enhance performance and produce more stable benefits in this moderate LR regime. A more systematic recipe for the strategy combinations can be a promising direction for future work.

**Discussion: Why is this Combination Under-Explored?** The CDMA strategy is straightforward and effective, which raises the question of why it has been underexplored. A possible explanation is that prior work has focused on an aggressive LR decay regime, which has obscured the discovery of alternative approaches. Confirmed by prior work (Li et al., 2025b), this aggressive decay regime is close to optimal for the standard uniform data scenario, and there is a clear trend favoring a near-zero ending LR, as shown in Figure 5. However, the optimal regime for uniform data is not necessarily optimal for other settings. Prior work focusing on curriculum design mostly adopts cosine annealing schedules, within this aggressive decay regime, and has consequently reported marginal or disappointing results (Zhang et al., 2025). The negative results caused by these compounding factors can prevent more progress on curriculum-based pretraining. Hence, our proposed optimization space

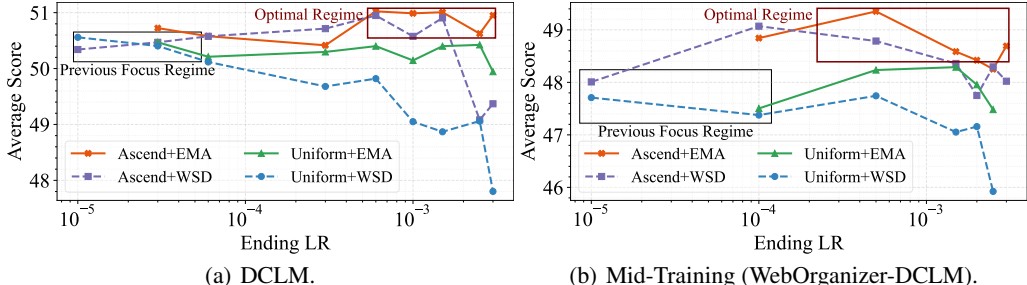

(a) DCLM.  (b) Mid-Training (WebOrganizer-DCLM).

Figure 5: This figure compares various training strategies, identifying a high-performing and previously underexplored **Optimal Regime** where moderate learning rate (LR) decay, weight averaging, and curriculum learning produce synergistic advantages. We run experiments on both Uniform (uniformly ordered data) and Ascend (training data arranged by ascending DCLM scores) data schedules. For both schedules, we conduct an ablation on the ending learning rates of WSD schedules, ranging from $1 \times 10^{-5}$ to $1 \times 10^{-3}$, representing aggressive to moderate LR decay. We denote strategies applying weight averaging as *EMA*, which compute the final model checkpoint via an EMA of the last six checkpoints, and denote those strategies without weight averaging as *WSD*. We measure performance by the average downstream task score (as in Table 1). This newly identified regime contrasts with the **Previous Focus Regime**, which represents common practices without a data curriculum or weight averaging, and with an ending LR between $1 \times 10^{-5}$ and $1 \times 10^{-4}$. This range is typical in prior work, which often uses an ending LR of one-tenth of a peak LR (on the scale of $\times 10^{-4}$) (Dubey et al., 2024; DeepSeek-AI et al., 2024) or fixes the ending LR around $10^{-5}$ (Li et al., 2024; 2025b). This observation also holds for mid-training settings.

involving the co-design of LR schedules, data curricula, and model averaging strategies is still underexplored

**Ablation Study.** We conduct ablation studies to verify the robustness of our method. Our approach generalizes effectively when evaluated with an alternative quality metric (PreSelect) and a different, unfiltered pretraining dataset (WebOrganizer) (SHUM et al., 2025; Wettig et al., 2025). Full experimental details are presented in Section C.3.

## 4 A Theoretical Demonstration

As we reported and discussed above, the benefit of curriculum learning emerges when we apply a weight averaging method instead of significantly decaying the learning rate, such as Cosine or WSD schedules. In the following, we present a simple theoretical model that recovers the above empirical insight. The main proof of this section can be found in Section F.

**Problem Setup.** We consider a quadratic loss function $\mathcal{L}(\boldsymbol{w}) = \frac{1}{2}\|\boldsymbol{w} - \boldsymbol{w}^*\|_2^2$, where $\boldsymbol{w} = (w_1, w_2) \in \mathbb{R}^2$ represents the trainable parameter, and $\boldsymbol{w}^*$ denotes the ground truth, which is set to the origin $(0, 0)$. We use Stochastic Gradient Descent (SGD) to optimize this problem. We initialize the parameter as $\boldsymbol{w}_0 = (L, 0)$. At the $t$-th iteration, the SGD update is $\boldsymbol{w}_t = \boldsymbol{w}_{t-1} - \eta_t \nabla \ell_t(\boldsymbol{w}_{t-1})$, where $\eta_t$ is the learning rate, $\ell_t(\boldsymbol{w}) := \|\boldsymbol{w} - \boldsymbol{x}_t\|_2^2$ is the loss function for a random data point $\boldsymbol{x}_t$ sampled from a given dataset $\mathcal{D} = \{\boldsymbol{x}^{(1)}, \boldsymbol{x}^{(2)}, \ldots, \boldsymbol{x}^{(M)}\}$. Note that the learning rate $\eta_t$ may vary over time, and we denote by $E := \{\eta_1, \eta_2, \ldots, \eta_M\}$ the learning rate schedule. Finally, let $\mathcal{W}_{M;E}$ be the distribution of $\boldsymbol{w}_M$, where the randomness within $\boldsymbol{w}_M$ comes from the random draw of the distribution in SGD. We define the expected loss as $\bar{\mathcal{L}}(M; E) := \mathbb{E}_{\boldsymbol{w} \sim \mathcal{W}_{M;E}} [\mathcal{L}(\boldsymbol{w})]$.

In the following, we consider the training dataset $\mathcal{D}$, which consists of $M$ different data points with varying data qualities. Specifically, the data point $\boldsymbol{x}^{(i)} = (x_1^{(i)}, x_2^{(i)})$ satisfies that $x_1^{(i)} = (i - 1)d$ and $x_2^{(i)} \sim \text{Uniform}(-L, L)$, where $d = L/M$. Intuitively, $\boldsymbol{x}^{(i)}$ provides a signal in the first dimension and introduces noise in the second dimension. Next, we consider two sampling strategies for each iteration of SGD: (1) **Uniform Sampling:** We sample one data point uniformly from $\mathcal{D}$; (2) **Ascending Data-Ordering:** we sample one data point from $\mathcal{D}$ in an ascending order of data quality, measured by the signal magnitude. In other word, in $t$-th iteration, $\boldsymbol{x}_t = \boldsymbol{x}^{(M-t+1)} \in \mathcal{D}$. See Figure 6 for a visualization of optimization trajectories.

**Uniform Sampling + Any Learning Rate Schedule.** In our setup, SGD acts as an exponential averaging of the current parameter and the sampled data point: $\boldsymbol{w}_t = \boldsymbol{w}_{t-1} - \eta_t(\boldsymbol{w}_{t-1} - \boldsymbol{w}^*) = (1 - \eta_t)\boldsymbol{w}_{t-1} + \eta_t \boldsymbol{w}^*$. If we uniformly sample data points with no ordering, then on the x-axis, the parameter approximately oscillates from 0 to $(m - 1)d$ with a large variance. We can prove that given any data schedule $E$ starting with some $\eta_1 \leq 1$, the expected loss for uniformly sampling

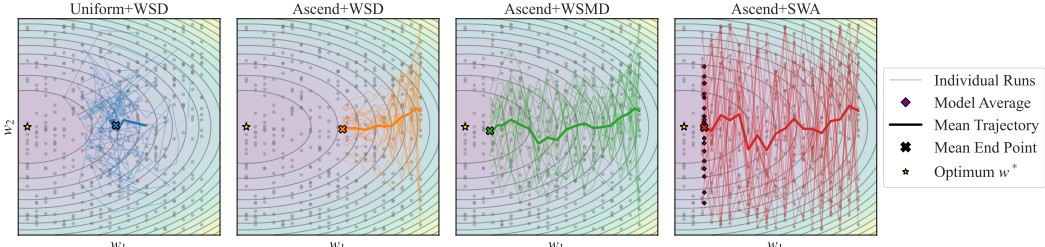

Figure 6: Visualization of the simulation experiments of the theoretical example. The mean trajectory is averaged over $R = 20$ runs. The yellow star marks the global optimal, and $w_1$ represents a signal direction and $w_2$ represents a noise direction. The data samples are distributed evenly along the signal direction and randomly located along the noise direction. *Ascend+WSMD* and *Ascend+SWA* win by sufficient progress along signal direction; *Uniform+WSD* fails for inconsistent signal and thus large variance along signal direction; *Ascend+WSD* fails for early-decay, resulting in insufficient update along the signal direction.

SGD has a lower bound

$$\min_E \bar{\mathcal{L}}(M; E) = \Omega(L^2). \tag{1}$$

This lower bound is derived from the loss on the x-axis. When we apply the uniform sampling, SGD cannot get enough signal towards the right direction; instead, the SGD optimizer will approach the mean of $\{x_1^{(1)}, x_1^{(2)}, \ldots, x_1^{(M)}\}$ in expectation.

**Ascending Data-Ordering + Practical WSD Schedule.** Next, we sample data in an ascending order from $\boldsymbol{x}^{(M)}$ to $\boldsymbol{x}^{(1)}$, using a WSD learning rate schedule $\tilde{E}$ with $\eta_t = \frac{1}{2}$ for $1 \leq t \leq \lfloor 0.9M \rfloor + 1$, and $\eta_t = \frac{1}{T - (M - T_0)}$ for $\lfloor 0.9M \rfloor + 2 \leq t \leq M$, where $T_0 = M - \lfloor 0.9M \rfloor$. This means we apply decay during the last 10% of the total iterations, following common practice. We then show that for this learning rate schedule $\tilde{E}$, the expected loss still cannot be lower than $\Omega(L^2)$:

$$\bar{\mathcal{L}}(M; \tilde{E}) = \Theta(L^2). \tag{2}$$

**Ascending Data-Ordering + WSMD Schedule.** In the above, we show that even using an ascending data-ordering, the loss lower bound does not improve if we decay too much in the learning rate schedule. Next, we show that a Warmup-Stable-Moderate Decay (WSMD) schedule with less decay and a larger ending learning rate can better utilize the ascending data-ordering and get a smaller loss. Specifically, we change $T_0$ from $\Theta(M)$ to $\Theta(M^{\frac{2}{3}})$ in the above WSD schedule. The resulting WSMD schedule is denoted by $E^*$ and it can break the above $\Omega(L^2)$ barrier:

$$\bar{\mathcal{L}}(M; E^*) = \Theta(M^{-\frac{2}{3}} L^2). \tag{3}$$

**Ascending Data-Ordering + Stochastic Weight Averaging (SWA).** Despite the failure of the practical WSD learning rate schedule, we demonstrate that with a constant learning rate, SWA is also able to break the above $\Omega(L^2)$ barrier. The reason is that: (1) First, along the x-axis, with a constant learning rate, the updated parameter gets a larger gradient accumulation towards the ground truth compared with the practical WSD with $10\%$ decay, which is too much to get enough loss reduction. (2) Second, the SWA allows appropriate averaging along the y-axis and results in noise reduction, thus leading to a smaller loss, as the WSMD schedule does.

**Theorem 4.1.** *Given a learning rate $\eta_0 \leq 1$, the parameter derived by averaging over the last $n$ weights $\bar{\boldsymbol{w}}_M = \frac{1}{n} \sum_{t=0}^{n-1} \boldsymbol{w}_{M-t}$, where $n = \Theta(M^{\frac{2}{3}})$, leads to the expected loss*

$$\mathbb{E}[\mathcal{L}(\bar{\boldsymbol{w}}_M)] = \tilde{O}(M^{-\frac{2}{3}} L^2),$$

*where $\tilde{O}(\cdot)$ hides log factors and constants independent of $L$ and $M$.*

## 5 CONCLUSION

In this work, we investigate the interaction between data schedules and learning-rate (LR) schedules in language model pretraining and identify a fundamental tension between curriculum learning and conventional LR-decay strategies. We propose to use a combination of moderate decay and model averaging to solve the mismatch, uncovering a previously underexplored optimization regime that better aligns curriculum-based data ordering with LR decay.

ACKNOWLEDGMENT

We would like to thank Kaiyue Wen, Shengqi Chen, Yanzheng Cai, and Jiping Yu for their insightful comments and feedback. This work is supported by the National Key R&D Project under Grant Number 2025YFB3003704 and the National Natural Science Foundation of China under Grant Number 62495062.

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

## A  RELATED WORK

**Curriculum Learning in LLM Pretraining.**  While data quality is known to be critical, leveraging it through fine-grained, instance-level curricula has shown limited success and has not been validated at a substantial scale (Dai et al., 2025; Zhang et al., 2025; Wettig et al., 2024; Campos, 2021). Prior studies have reported only marginal gains, sometimes observing benefits from counterintuitive data orders without a clear underlying mechanism (Wettig et al., 2024). Other works either lack sufficient validation (Campos, 2021; Kim & Lee, 2024) or, finding simple curricula ineffective, propose more complex strategies like data folding (Zhang et al., 2025; Dai et al., 2025). However, we find that the benefits of such complex strategies are often confined to smaller-scale experiments with low learning rates and can degrade at larger scales and under a high LR regime (Section E.1). A common thread in these attempts is the use of standard cosine learning rate schedules. We demonstrate that decayed LR prevents the model from effectively learning from the high-quality data introduced late in the training process, showing that a more moderate decay is required to unlock the curriculum's full potential. A more detailed discussion can refer to Section E.1.

**Learning Rate Schedules.**  The learning rate (LR) schedule is a crucial hyperparameter in model pretraining.  Traditional LR schedules like cosine decay are the most widely used in LLM pretraining (Loshchilov & Hutter, 2017; Dubey et al., 2024; Li et al., 2024).  More recent schedules like Warmup-Stable-Decay (WSD) have demonstrated strong performance and are suitable for midtraining resumption paradigms (Hu et al., 2024; Hägele et al., 2024). Loss curve scaling laws suggest that they possess an optimal shape for a given peak LR (Tissue et al., 2024; Luo et al., 2025; Li et al., 2025a). A prevailing finding for training with standard uniform data ordering is that an optimal or near-optimal LR schedule should decay to a value close to zero (Li et al., 2025b; OLMo et al., 2025; Li et al., 2024).  Our work challenges this convention in the context of curriculum-based pretraining. We find that for a data curriculum, such an aggressive decay has negative effects on the data schedule.  Instead, a moderate decay that maintains a higher learning rate during the high-quality data phase proves superior, especially when using model averaging. Unless otherwise specified, we discuss the LR schedules with a warmup phase. For example, a constant LR schedule refers to a linear warmup followed by a constant LR.

**Model Averaging.**  Model averaging, which combines parameters from multiple checkpoints to produce a final, improved model, is a well-established technique for improving generalization (Izmailov et al., 2018; Jin et al., 2023; Yang et al., 2024b). It has been successfully applied in LLM pretraining to accelerate convergence and boost performance, and was reportedly used in training prominent models like Llama 3 (Kaddour, 2022; Li et al., 2025c; Dubey et al., 2024).  Prior work has explored combining model averaging with decay-free LR schedules for training on uniformly ordered data and in the mid-training setting (Li et al., 2025c; Tian et al., 2025). However, Li et al. (2025c) finds that the performance of weight averaging is merely comparable to standard LR decay, and Tian et al. (2025) attempts to improve the results through a weighting strategy derived from the LR schedule (Discussed in Sections B and C.2). Unlike this prior work, which focused on uniform data ordering, our research is the first to investigate the interaction between model averaging and a quality-based data curriculum. We find that these two strategies have a strong synergistic relationship, as model averaging provides the necessary training stability to learn from high-quality data with a high, moderately decaying learning rate.

## B  PRELIMINARY

**Learning Rate Schedule.**  We consider two primary types of learning rate (LR) schedules in addition to a constant LR schedule. The commonly used cosine schedule defines the LR at step $t$ as $\eta(t) = \eta_0 \left( \frac{1+\alpha}{2} + \frac{1-\alpha}{2} \cos\left( \frac{\pi t}{T} \right) \right)$, where $\eta_0$ is the peak LR, $T$ is the total number of training steps, and $\alpha$ is the ratio of the final LR to the peak LR. A more recent alternative is the Warmup-Stable-Decay (WSD) schedule, which consists of a linear warmup phase, a stable phase with a constant LR $\eta_0$, and a final decay phase. For the decay phase, we use the *1-sqrt* function, where the LR is given by $\eta(t) = \eta_0 \left( 1 - \sqrt{r(t)} \right) + \eta_T \sqrt{r(t)}$, with $r(t) = \frac{t - t_{\text{decay}}}{T - t_{\text{decay}}}$ representing the progress through the decay period, which begins at step $t_{\text{decay}}$. Further details on our schedule choices are discussed in Section C.1.

**Model Averaging.**  Model averaging computes a weighted average of several model checkpoints to produce a single, final model. We consider three common strategies. Suppose we have $N$ check-

points, $M_1, \ldots, M_N$, typically the last $N$ checkpoints collected at fixed intervals, corresponding to steps $t_1, \ldots, t_N$. We focus on the model averaging strategies discussed in Li et al. (2025c) summarized as follows. *Simple Moving Average (SMA)* (Izmailov et al., 2018) applies a uniform weight to all these checkpoints: $M_{\text{avg}} = \frac{1}{N} \sum_{i=1}^{N} M_i$. *Exponential Moving Average (EMA)* assigns exponentially decaying weights. EMA is defined recursively as $M_{\text{avg}}^{(i)} = \alpha M_i + (1 - \alpha) M_{\text{avg}}^{(i-1)}$, with the base case $M_{\text{avg}}^{(1)} = M_1$. The hyperparameter $\alpha \in (0, 1]$ controls the decay rate; a larger $\alpha$ assigns greater weight to the most recent checkpoint. *Weighted Moving Average (WMA)* uses a predefined set of normalized weights $w_1, \ldots, w_N$ (where $\sum w_i = 1$) to compute the final model as $M_{\text{avg}} = \sum_{i=1}^{N} w_i M_i$. Prior work (Tian et al., 2025) derives these weights from the learning rate schedule, where each weight is proportional to the drop in learning rate between checkpoints: $w_i \propto \eta(t_i) - \eta(t_{i+1})$ for $i < N$, and $w_N \propto \eta(t_N)$. The weight computation procedure for WMA is detailed in Section C.2.

**Data Scoring.** Raw web data must pass through a processing pipeline before it is used for pretraining. The raw data first goes through heuristic filtering rules. Then in the model-based filtering phase, a scorer model assigns a quality score to each data sample. For example, DCLM Baseline dataset uses scores from a fasttext model (Joulin et al., 2017) measuring similarity to high-quality sources like OpenHermes 2.5 (Teknium, 2023) and top posts from the ELI5 subreddit. Another approach, PreSelect (SHUM et al., 2025), scores data based on its similarity to downstream tasks. Typically, these scores are used to filter the dataset by removing samples below a certain quality threshold. In contrast, our work does not use these scores to discard data; instead, we use these quality scores to define the data ordering for curriculum learning.

## C  EXPERIMENTS

### C.1  EXPERIMENTAL SETTINGS

This section details our experimental setup, which is grounded at a substantial scale for academic research. We describe our pretraining and evaluation procedures, justify our learning rate schedule choices, and outline a practical mid-training configuration.

**Pretraining Settings.**  Our experiments are conducted at a substantial scale for academic validation, grounded in the DataComps-LM (DCLM) framework (Li et al., 2024) at the `1B-1x` level. We use Qwen2.5-1.5B model (Yang et al., 2024a) for pretraining experiments, which is a modern architecture incorporating SwiGLU activation functions and Grouped-Query Attention (GQA). The training dataset consists of a 30B token subset sample from the first shard of the DCLM-Baseline dataset (Li et al., 2024). For our data curricula, we sort data in ascending order based on DCLM fasttext scores; a reverse curriculum, sorting in descending order, is also used for comparison. After moderate tuning, we set the peak learning rate to $3 \times 10^{-3}$, with a sequence length of 4096 and a batch size of 512, which we found provides a good trade-off between throughput and stability. For LR decay schedules, we set the final learning rate to $1 \times 10^{-5}$, aligning with optimal settings found in prior work (Li et al., 2024; Luo et al., 2025; Li et al., 2025b). To ensure reproducibility, we provide a detailed list of hyperparameters in Table 3.

Table 3: Model and optimizer hyperparameters for our Qwen2.5-1.5B experiments.

| Hyperparameter | Value |
|---|---|
| *Model Configuration* | |
| Sequence Length | 4096 |
| Hidden Size | 1536 |
| FFN Intermediate Size | 8960 |
| Number of Layers | 28 |
| Number of Attention Heads | 12 |
| Number of Key-Value Heads (GQA) | 2 |
| Vocabulary Size | 151936 |
| *Optimizer Configuration* | |
| Optimizer | AdamW |
| Weight Decay | 0.1 |
| Adam $\beta_1$ | 0.9 |
| Adam $\beta_2$ | 0.95 |
| Adam $\epsilon$ | $1.0 \times 10^{-8}$ |
| Gradient Clipping | 1.0 |

**Evaluation Settings.**  To ensure a robust comparison between methods, we track validation loss during training and evaluate final performance on a comprehensive suite of downstream tasks. Because the data distribution shifts during a curriculum, we created a dedicated high-quality validation set to provide a consistent measure of progress. This set consists of 100,000 of the highest-scoring documents from a partition of the DCLM-Baseline dataset that is disjoint from our training data. For downstream evaluation, we use the OLMES benchmark (Gu et al., 2025a), reporting performance on MMLU (Hendrycks et al., 2021), ARC-easy/challenge (Clark et al., 2018), CommonSenseQA (CSQA) (Talmor et al., 2019), OpenBookQA (Mihaylov et al., 2018), PIQA (Bisk et al., 2020), Social IQA (Sap et al., 2019), and WinoGrande (Sakaguchi et al., 2020), covering world knowledge, common sense, and reasoning capabilities. Among these, we designate MMLU, ARC, and CSQA as *Core* benchmarks, as recent work suggests they have a higher signal-to-noise ratio for distinguishing model performance (Heineman et al., 2025). Furthermore, as shown in Figure 7, the average downstream scores exhibit a strong correlation with validation loss.

**LR Schedule Choice Ablation.**  The decay phase of the WSD schedule can be implemented with various functions. Following prior work (Hägele et al., 2024; Luo et al., 2025), we compared the

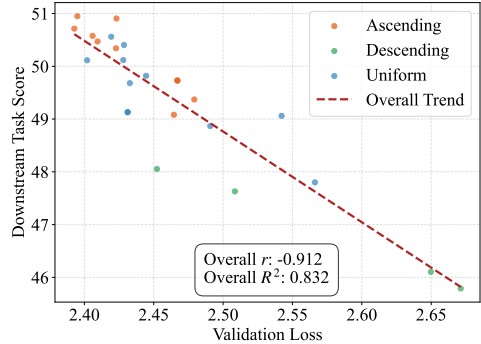 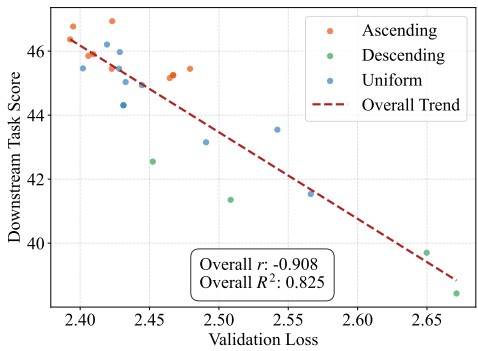

(a) *Average* downstream score over validation loss.  (b) *Core* downstream score over validation loss.

Figure 7: Downstream task scores and validation losses show high correlation according to the Pearson correlation coefficient ($r$) and R-square value ($R^2$). *Average* is the average score of the total 8 downstream t,asks and *Core* is the average score of the first 4 downstream tasks (MMLU, ARC-c/e, CSQA) in Tables 1 and 2.

*1-sqrt* decay function, $\eta(t) = \eta_0 \left(1 - \sqrt{r(t)}\right) + \eta_T \sqrt{r(t)}$, and the *sqrt-cube* function, $\eta(t) = \eta_0 \left(1 - r(t)\right)^{1.5}$. Both decay functions outperform simpler alternatives like linear decay, and as shown in Table 4, they produce strong and comparable results. We adopt the *1-sqrt* function for our main experiments due to its wide adoption in recent literature (Hägele et al., 2024; Tian et al., 2025). Unless otherwise specified, our experiments use a decay phase ratio of approximately between 15% and 20% of total training steps, consistent with optimal ratios reported in prior work (Hägele et al., 2024; Hu et al., 2024).

Table 4: Models trained under WSD schedules under *1-sqrt* and *sqrt-cube* decay functions produce similar results.

| Dataset | Schedule | MMLU | ARC-c | ARC-e | CSQA | OBQA | PIQA | SIQA | Wino. | Avg. |
|---------|----------|------|-------|-------|------|------|------|------|-------|------|
| Random | 1-sqrt | 26.60 | 27.42 | 42.28 | 42.26 | 37.20 | 67.85 | 42.02 | 51.30 | 42.12 |
| Filtered | 1-sqrt | 26.97 | 30.10 | 44.04 | 44.72 | 36.20 | 69.15 | 42.58 | 51.78 | 43.19 |
| Random | sqrt-cube | 26.62 | 27.42 | 42.28 | 42.42 | 35.00 | 68.28 | 43.65 | 51.70 | 42.17 |
| Filtered | sqrt-cube | 26.70 | 31.10 | 44.04 | 42.59 | 36.60 | 68.44 | 42.89 | 52.17 | 43.07 |

**The Mid-Training Experiment Settings.** We model mid-training, an emerging and practical pre-training paradigm (Yang et al., 2025; OLMo et al., 2025; Hu et al., 2024), with a two-phase setup. The initial stable-LR phase uses 29B tokens from the WebOrganizer dataset (Wettig et al., 2025) as lower-quality data, while the subsequent decay phase uses 5B tokens from the higher-quality DCLM-Baseline dataset (Li et al., 2024). The WebOrganizer data has not undergone model-based filtering, whereas the DCLM-Baseline data represents the top 10% of its source, which has a distribution roughly similar to the WebOrganizer data. The LR decays to $1 \times 10^{-5}$ during the second, high-quality phase.

## C.2 PRACTICAL IMPLEMENTATION DETAILS

**Weight Computation for WMA.** The computation of Weighted Moving Average (WMA) follows Tian et al. (2025), where weights are derived from a hypothetical learning rate schedule. For our experiments, we use a WSD schedule with a *1-sqrt* decay function and a final LR set to 5% of the peak LR. Given normalized LR values $\eta_1, \ldots, \eta_N$ at each checkpoint (with $\eta_1 = 1$), the weights are calculated as the drop in learning rate between steps: $w_i = \eta_i - \eta_{i+1}$ for $i < N$, and $w_N = \eta_N$. Suppose $M_0, M_1, \ldots, M_N$ are the model parameters of the last $N + 1$ checkpoints, and assume $M_k = M_0 + \sum_{j=1}^{k} g_j$, where $g_j$ represents the total parameter update between checkpoint $j - 1$ and $j$. This weighting strategy ensures that the averaged model is equivalent to re-weighting each

update:

$$\sum_{i=1}^{N} w_i M_i = \sum_{i=1}^{N-1} (\eta_i - \eta_{i+1}) \left( M_0 + \sum_{j=1}^{i} g_j \right) + \eta_N \left( M_0 + \sum_{j=1}^{N} g_j \right) = M_0 + \sum_{i=1}^{N} \eta_i g_i$$

This formulation shows how the weighted average effectively re-weights the parameter updates $g_i$ by their corresponding normalized learning rates $\eta_i$, thus simulating the effect of an LR schedule through averaging. Notably, for the *1-sqrt* decay function, this method results in a set of monotonically decreasing weights, as shown in Table 5.

Table 5: Model Checkpoint Weights. Index $-k$ corresponds to the last $k$-th checkpoint.

| Checkpoint Index | -6 | -5 | -4 | -3 | -2 | -1 |
|---|---|---|---|---|---|---|
| Weight | 0.4249 | 0.1760 | 0.1350 | 0.1138 | 0.1003 | 0.0500 |

**Practical Implementation of CMA (and CDMA).** Our Curriculum Model Averaging (CMA) approach consists of a three-stage pipeline, as detailed in Algorithm 1. First, in an offline *data scheduling* stage, the entire training dataset is sorted in ascending order based on a pre-computed quality score. This large-scale sorting is performed efficiently as a one-time process using Apache Spark. A significant practical advantage is that these quality scores (e.g., DCLM fasttext (Li et al., 2024) or PreSelect (SHUM et al., 2025)) are often already generated during data preprocessing for filtering purposes, allowing our method to be integrated into existing pipelines without requiring an additional, costly scoring step. Second, during the *training* stage, we employ a simple warmup-constant LR schedule, forgoing any LR decay. Finally, to ensure the stability of the final parameters, we perform *model averaging* on the last several checkpoints saved during the constant-LR phase. We primarily use Exponential Moving Average (EMA), which assigns higher weights to more recent checkpoints trained on higher-quality data, thereby smoothing parameter variance while emphasizing high-quality signals. For CDMA, the only difference is that instead of entirely forgoing LR decay, we use a moderate decay (e.g., to a final LR of $1/5$ to $1/2$ of the peak LR) and then apply weight averaging. Further exploration of optimal ending LRs and the scaling properties of this combined strategy are promising directions for future work.

## C.3 ABLATION STUDIES

In this section, we validate the robustness of our findings. We conduct ablation studies to demonstrate that our approach generalizes across a different quality metric (PreSelect) and a different, unfiltered pretraining dataset (WebOrganizer) (SHUM et al., 2025; Wettig et al., 2025).

**Ablation on Quality Metric.** To test if our approach generalizes to other quality metrics, we conducted experiments using PreSelect scores (SHUM et al., 2025) to order the data. The results in Table 6 show that when using an ascending order of PreSelect scores, both CMA and CDMA outperform the standard WSD baseline. However, we note that ordering by PreSelect scores yielded slightly lower average performance than ordering by DCLM scores in our setup. Moreover, as shown in Figure 8(a), although the ascending-order training shows an overall faster convergence in the latter phase of the training process, the final validation loss goes above the uniform-ordering data training. We conjecture this may be due to an inconsistency, as our base dataset was originally filtered using DCLM scores before being re-sorted by PreSelect scores, while the targeted validation set consists of samples with the highest DCLM scores.

**Ablation on Pretraining Dataset.** To verify that our method is applicable to broader data distributions, we conducted experiments on the WebOrganizer dataset (Wettig et al., 2025) alone, which has not undergone model-based filtering. As shown in Table 7, combining model averaging with a data curriculum again produces superior results compared to a standard LR decay approach. Notably, in this setting, model averaging without any decay showed a larger benefit than model averaging with moderate decay, possibly indicating that a high ending learning rate is particularly beneficial when high-quality data is extremely sparse.

---

**Algorithm 1** Curriculum Model Averaging (CMA)

---

1: **Input:** Unsorted dataset $D$, quality scoring function $Q(\cdot)$, training steps $T$, warmup steps $T_w$, peak learning rate $\eta_{peak}$, number of checkpoints to average $k$, averaging decay hyperparameter $\alpha$, checkpointing interval $s$.
2: **Output:** Final model parameters $\bar{\theta}_{\text{final}}$.

3: # Stage 1: Data Scheduling
4: Sort dataset $D$ to create $D_{sorted}$ where for any samples $x_i, x_j$, if $i < j$, then $Q(x_i) \leq Q(x_j)$.

5: # Stage 2: Warmup-Constant LR Training
6: Initialize model parameters $\theta_0$.
7: **for** $t = 1$ **to** $T$ **do**
8:     **if** $t \leq T_w$ **then**
9:         $\eta_t \leftarrow \eta_{peak} \cdot (t/T_w)$                                         ▷ Linear warmup
10:     **else**
11:         $\eta_t \leftarrow \eta_{peak}$                                                ▷ Constant LR
12:     **end if**
13:     Fetch next data batch $B_t$ from $D_{sorted}$.
14:     $\theta_t \leftarrow \text{OptimizerUpdate}(\theta_{t-1}, \eta_t, B_t)$                           ▷ e.g., Adam
15:     **if** $t \in \{T - (k-1)s, \ldots, T - s, T\}$ **then**
16:         Save checkpoint $\theta_t$.
17:     **end if**
18: **end for**

19: # Stage 3: Model Averaging, e.g., EMA or SMA.
20: Let the set of saved checkpoints be $\{\theta_{T-is}\}_{i=0}^{k-1}$.
21: EMA: $\bar{\theta}_{\text{final}} \leftarrow \dfrac{\sum_{i=0}^{k-1} \alpha^i \theta_{T-is}}{\sum_{i=0}^{k-1} \alpha^i}$                        ▷ SMA: $\bar{\theta}_{\text{final}} \leftarrow \dfrac{\sum_{i=0}^{k-1} \theta_{T-is}}{k}$
22: **return** $\bar{\theta}_{\text{final}}$

---

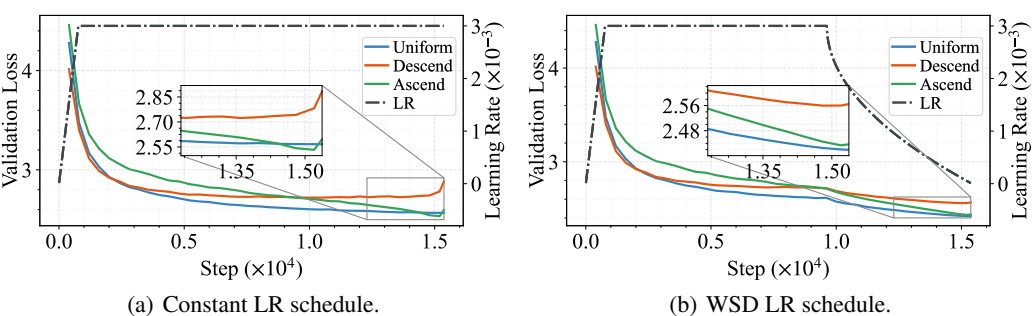

(a) Constant LR schedule.                               (b) WSD LR schedule.

Figure 8: The benefits of a data curriculum using PreSelect scores also diminish. We show the validation loss curves for constant and WSD LR schedules under different data schedules, including uniform, ascending, and descending orders by PreSelect scores. Overall, the ascending curriculum outperforms the uniform baseline under a constant schedule, but cannot match it under the WSD LR schedule. The final validation loss of the data curriculum is higher than that of the uniform-ordering baseline, likely because the score metrics are not perfectly targeted to the validation set.

# D   Large-Scale Continual Pretraining with Multi-Domain Curriculum

Practical pretraining scenarios typically involve datasets spanning multiple domains, including natural language, code, and mathematical content (OLMo et al., 2025; Yiwen et al., 2025; Allal et al., 2025). To integrate curriculum learning into this multi-domain setting, we propose an adaptation of our CMA/CDMA framework that accommodates heterogeneous data sources. Due to computational constraints and practical deployment considerations, we evaluate our approach in a continual

Table 6: Downstream performance for experiments with PreSelect score ascending data. Our proposed methods (using WA) are highlighted in gray. **WA**: Weight Averaging (EMA: Exponential, SMA: Simple). **LRS**: Learning Rate Schedule (WSD: WSD with decay to $1 \times 10^{-5}$, Const: Constant LR, WSMD: WSD with moderate decay to $1 \times 10^{-3}$). **Core**: Average score on the first four, high signal-to-noise tasks (MMLU, ARC-c, ARC-e, CSQA). Both Core and Avg. scores are annotated with a subscript indicating the performance change relative to the baseline (first row). Subscripts in **bold green** indicate an improvement of $\geq 0.5$, light green an improvement of $> 0$, and red a decrease.

| WA | Order | LRS | MMLU | ARC-c | ARC-e | CSQA | Core | OBQA | PIQA | SIQA | Wino. | Avg. |
|---|---|---|---|---|---|---|---|---|---|---|---|---|
| ✗ | Ascend | WSD | 31.12 | 35.79 | 57.89 | 48.81 | 43.40 | 41.00 | 71.82 | 46.21 | 58.41 | 48.88 |
| EMA | Ascend | Const | 31.85 | 37.46 | 61.05 | 49.39 | $44.94_{+1.54}$ | 38.40 | 70.51 | 45.34 | 55.33 | $48.67_{-0.21}$ |
| EMA | Ascend | WSMD | 31.98 | 39.80 | 61.93 | 49.39 | $45.77_{+2.37}$ | 39.60 | 70.78 | 45.60 | 55.96 | $49.38_{+0.50}$ |
| SMA | Ascend | WSMD | 31.99 | 39.46 | 62.11 | 50.04 | $45.90_{+2.50}$ | 39.40 | 71.06 | 46.06 | 55.96 | $49.51_{+0.63}$ |

Table 7: Downstream performance for experiments on WebOrganizer dataset (Wettig et al., 2025). Our proposed methods (using WA) are highlighted in gray. **WA**: Weight Averaging (EMA: Exponential, SMA: Simple). **LRS**: Learning Rate Schedule (WSD: WSD with decay to $1 \times 10^{-5}$, Const: Constant LR, WSMD: WSD with moderate decay to $1 \times 10^{-3}$). **Core**: Average score on the first four, high signal-to-noise tasks (MMLU, ARC-c, ARC-e, CSQA). Both Core and Avg. scores are annotated with a subscript indicating the performance change relative to the baseline (first row). Performance changes are color-coded: **bold green** ($\geq 0.5$ improvement), light green ($> 0$ improvement), and red (decrease).

| WA | Order | LRS | MMLU | ARC-c | ARC-e | CSQA | Core | OBQA | PIQA | SIQA | Wino. | Avg. |
|---|---|---|---|---|---|---|---|---|---|---|---|---|
| ✗ | Uniform | WSD | 28.92 | 34.45 | 47.72 | 47.83 | 39.73 | 36.60 | 72.03 | 43.76 | 56.67 | 46.00 |
| | Ascend | WSD | 29.09 | 32.78 | 51.58 | 47.42 | $40.22_{+0.49}$ | 38.00 | 72.14 | 44.73 | 55.09 | $46.35_{+0.35}$ |
| EMA | Uniform | WSMD | 28.28 | 34.11 | 47.89 | 48.81 | $39.77_{+0.04}$ | 39.20 | 71.65 | 43.76 | 55.56 | $46.16_{+0.16}$ |
| EMA | Ascend | WSMD | 28.56 | 31.10 | 50.88 | 48.89 | $39.86_{+0.13}$ | 39.60 | 71.44 | 44.11 | 56.75 | $46.42_{+0.42}$ |
| EMA | Uniform | Const | 28.03 | 33.44 | 47.72 | 47.42 | $39.15_{-0.58}$ | 40.60 | 70.78 | 43.65 | 55.72 | $45.92_{-0.08}$ |
| EMA | Ascend | Const | 29.32 | 33.11 | 55.09 | 48.89 | $41.60_{+1.87}$ | 38.40 | 71.00 | 45.29 | 55.41 | $47.06_{+1.06}$ |

pretraining setup, where training resumes from base phase checkpoints and proceeds with our curriculum strategy. To validate the generalizability of our method, we scale to a 3.2B parameter model and conduct experiments with over 150B tokens during the continual phase, comparing against established baselines.

## D.1 MULTI-DOMAIN CURRICULUM LEARNING FRAMEWORK

Real-world pretraining datasets comprise diverse domains with distinct characteristics, making it challenging to define a unified quality metric across all data sources (OLMo et al., 2025; Yiwen et al., 2025; Allal et al., 2025). To address this limitation while preserving the benefits of curriculum learning, we introduce a multi-domain extension of our CMA/CDMA methodology.

Our approach employs a three-stage pipeline designed to maintain both within-domain curriculum ordering and stable cross-domain mixing ratios:

1. **Within-Domain Ranking**: Following data preprocessing and resampling (Wettig et al., 2025; Ye et al., 2025), we independently sort samples within each domain using domain-specific quality metrics. Lower ranks are assigned to lower-quality samples, establishing an ascending curriculum within each domain.

2. **Rank Rescaling**: We transform domain-specific rankings to a unified global scale. For a sample from domain $A$ with within-domain rank $r_A$, the rescaled global rank is computed as:

$$R_{\text{global}}(x_A) = r_A \times \frac{N_{\text{total}}}{N_A}$$

where $N_A$ represents the sample count in domain $A$, and $N_{\text{total}}$ denotes the aggregate sample count across all domains. This normalization ensures that the rankings of different domains are aligned in the same numerical metric.

---

**Algorithm 2** Multi-Domain Curriculum Construction

---

**Require:** Domain datasets $D_1, D_2, \ldots, D_k$ with domain-specific quality metrics
**Ensure:** Multi-domain curriculum dataset
1:  $N_{\text{total}} \leftarrow \sum_{i=1}^{k} |D_i|$          ▷ Compute total sample count
2: **for** each domain $D_i$ **do**
3:      Sort $D_i$ by domain-specific quality metric (ascending)     ▷ Within-domain ranking
4:      Assign ordinal ranks $r_i(x) \in [1, |D_i|]$ to each sample $x \in D_i$
5:      Compute rescaled ranks: $R(x) \leftarrow r_i(x) \times \frac{N_{\text{total}}}{|D_i|}$ for all $x \in D_i$
6: **end for**
7:  $U \leftarrow \bigcup_{i=1}^{k} D_i$                   ▷ Combine all domains
8: Sort $U$ by rescaled rank $R(x)$ in ascending order        ▷ Global interleaving
9: **return** sorted $U$

---

3. **Global Interleaving**: After rescaling, we merge all domain datasets and sort the combined collection by the computed global ranks in ascending order. This produces a globally-ordered curriculum that achieves three key properties:

   (1) Preservation of within-domain quality-based ordering

   (2) Proportional interleaving of samples across domains according to their mixing ratios

   (3) Stable domain mixture ratios maintained throughout training

This methodology ensures that higher-quality samples from all domains are prioritized during training while maintaining the intended domain distribution. The process can be efficiently implemented using distributed computing frameworks like Spark, making it practical for large-scale pretraining with heterogeneous data sources. Algorithm 2 formalizes this procedure, providing a principled approach for constructing multi-domain curricula that balance quality-based ordering with distributional requirements.

## D.2 EXPERIMENTAL SETUP

Table 8: Base Phase Data Mixture

| Dataset | Token Count (B) | Ratio |
|---|---|---|
| DCLM | 608.5 | 83.51% |
| Fineweb-C | 91.8 | 12.60% |
| Starcoder | 19.0 | 2.61% |
| MegaMath | 9.3 | 1.28% |
| **Total** | **728.7** | **100%** |

Table 9: Continual Phase Data Mixture with Top-k Selection

| Dataset | Token Count (B) | Count Ratio | Top-K Ratio | Score Metric |
|---|---|---|---|---|
| DCLM | 83.9 | 53.77% | 0.2 | fasttext score |
| Fineweb-C | 23.9 | 15.34% | 0.2 | fineweb score |
| Starcoder | 38.9 | 24.95% | 0.2 | max stars count |
| MegaMath | 9.3 | 5.94% | 0.4 | duplicate count |
| **Total** | **156.1** | **100%** | | |

To evaluate the effectiveness of our multi-domain curriculum learning approach, we conduct large-scale continual training experiments with a 3.2B parameter model and over 150B tokens. Our data spans four domains: deduplicated DCLM Baseline (Li et al., 2024), Fineweb-Edu-Chinese-V2.1 (Yu et al., 2025), StarCoder (Li et al., 2023), and MegaMath (Zhou et al., 2025).

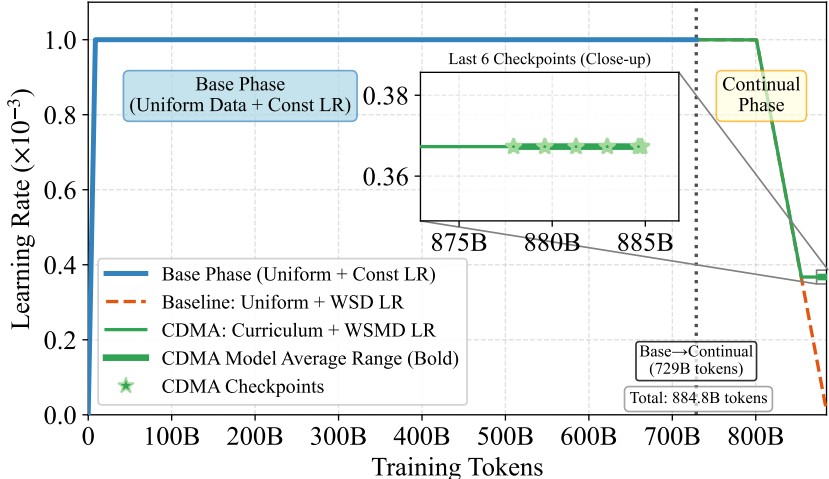

Figure 9: Learning rate and data schedules for continual pretraining with a 3.2B model. The base phase (729B tokens) uses uniform ordering, while the continual phase employs either uniform data ordering with decay to $1\times10^{-5}$ (baseline) or our multi-domain curriculum with decay to $3.67\times10^{-4}$ and EMA averaging of the last six checkpoints ($\alpha = 0.2$). Corresponding benchmark results are presented in Table 10.

Table 10: Benchmark results comparing baseline and our CDMA approach with multi-domain curriculum on 3.2B model after continual pretraining. Subscripts in **bold green** indicate an improvement of $\geq 0.5$, light green an improvement of $> 0$, and red a decrease.

| Method | GSM8K | MBPP | CEval | MMLU | ARC-C | ARC-E | CSQA | BoolQ | Core Avg |
|---|---|---|---|---|---|---|---|---|---|
| Baseline | 24.64 | **38.91** | 40.83 | 48.53 | 51.86 | **78.13** | 67.32 | 71.38 | 52.70 |
| Ours | **24.72**$_{+0.08}$ | 38.13$_{-0.78}$ | **42.12**$_{+1.29}$ | **48.97**$_{+0.44}$ | **55.25**$_{+3.39}$ | 77.95$_{-0.18}$ | **68.39**$_{+1.07}$ | **74.77**$_{+3.39}$ | **53.79**$_{+1.09}$ |

The base phase comprises approximately 730B tokens with uniform sampling, as detailed in Table 8. For the continual phase, we resume training from the final base phase checkpoint and employ a refined data mixture (Table 9). We use *Fineweb-C* to denote Fineweb-Edu-Chinese-V2.1 for brevity.

In the continual phase, we implement our multi-domain curriculum using domain-specific quality metrics: text quality scores (DCLM Baseline, Fineweb-Edu-Chinese-V2.1), GitHub star counts (StarCoder), and duplicate frequency[3] (MegaMath).

We employ the WSD learning rate schedule throughout training with a peak learning rate of $1\times10^{-3}$ and a batch size of 2048. The baseline method applies linear decay over the final 84B tokens to $1 \times 10^{-5}$ while maintaining uniform data ordering. In contrast, our CDMA strategy decays to $3.67 \times 10^{-4}$ and maintains this learning rate, followed by exponential moving average (EMA) over the last six checkpoints with $\alpha = 0.2$. Each checkpoint interval corresponds to approximately 1.67B tokens. The learning rate schedule and training phases are visualized in Figure 9.

### D.3 EXPERIMENTAL RESULTS

As shown in Table 10, our CDMA strategy with multi-domain curriculum consistently outperforms the baseline in large-scale continual pretraining. We observe significant improvements across multiple capability dimensions: general knowledge (MMLU: `+0.44`), commonsense reasoning (CSQA: `+1.07`), reading comprehension (BoolQ: `+3.39`), and complex reasoning (ARC-Challenge: `+3.39`). Notably, our approach also demonstrates strong performance on Chinese language understanding (CEval: `+1.29`), indicating generalization of our approach across different linguistic domains with appropriate quality metrics.

---

[3]We use duplicate count as a proxy for importance, under the assumption that frequently repeated samples may contain more valuable content.

Table 11: Performance comparison across learning rate schedules (Constant, Cosine) and data orders (Uniform, Ascend, Descend) on various downstream benchmarks. Results show that descending-order curricula perform worse than uniform ordering, and the performance gap narrows when LR decay is applied.

| LR Schedule | Order | MMLU | ARC-c | ARC-e | CSQA | OBQA | PIQA | SIQA | Wino. | Avg. |
|---|---|---|---|---|---|---|---|---|---|---|
| Constant | Uniform | 30.30 | 30.43 | 55.61 | 49.80 | 44.60 | 70.29 | 45.19 | 56.20 | 47.80 |
| | Ascend | 31.14 | 39.46 | 61.23 | 49.96 | 43.00 | 70.51 | 43.14 | 56.51 | 49.37 |
| | Descend | 29.43 | 33.11 | 45.96 | 45.21 | 41.00 | 69.86 | 44.98 | 56.75 | 45.79 |
| Cosine | Uniform | 30.49 | 38.13 | 59.47 | 49.14 | 42.20 | 71.87 | 45.19 | 56.51 | 49.13 |
| | Ascend | 30.80 | 39.80 | 59.12 | 51.27 | 42.60 | 71.55 | 45.65 | 57.06 | 49.73 |
| | Descend | 29.51 | 34.11 | 52.98 | 48.81 | 42.60 | 72.42 | 45.45 | 55.17 | 47.63 |

The overall core average improvement of 1.09% represents a meaningful advancement at this scale. However, we note limited gains in mathematical reasoning (GSM8K: $+0.08$) and a slight regression in coding capability (MBPP: $-0.78$). The relatively lower mixing ratios can limit the benefits in these domains. Besides, we hypothesize that these domains may benefit from more sophisticated quality metrics than the preliminary measures (e.g., GitHub star counts) employed in this work. This direction presents an opportunity for future research.

# E   DISCUSSION

## E.1   COMPARISON WITH RELATED WORK

In this section, we situate our findings in the context of prior research on curriculum learning and discuss our interpretation compared with previous work on the loss landscape. We discuss how our experimental setup differs from related approaches and how our findings elucidate the interplay between learning rate (LR) scheduling, data ordering, and data quality.

**Comparison with Prior Curriculum Learning Studies.**   Previous work on curriculum learning for large-scale language model (LLM) pretraining has often overlooked the interaction between data ordering and the learning rate schedule, typically adopting cosine LR schedules with low peak values on the order of $10^{-4}$ (Wettig et al., 2024; Dai et al., 2025; Zhang et al., 2025). For example, Wettig et al. (2024) reported a modest $0.6\%$ improvement with a low-to-high quality curriculum but also found a counterintuitive $0.5\%$ improvement with a reverse (descending-order) curriculum. Two factors may explain this paradox. First, in a low-peak-LR regime, LR decay can prematurely reduce the effective learning rate, narrowing the performance gap between curriculum and reverse-curriculum orders. Second, the data-quality metric itself may lack self-consistency, yielding noisy rankings.

In contrast, our experiments demonstrate a consistent performance drop for the reverse curriculum—especially under a constant LR schedule—and a smaller but still negative effect under schedules with LR decay (Table 11). These results indicate that our data-quality metric produces more self-consistent orderings and that LR decay indeed diminishes the benefits of a curriculum. While a direct quantitative comparison is not possible due to differences in scale and benchmarks, our best configuration (*Const+SMA*) achieves a relative improvement of over 2.7% compared to a comparable baseline (*Cos+Uniform*), demonstrating a substantially stronger effect than previously reported.

**Limitations of the Data Folding Strategy.**   To address the limited gains of vanilla data curricula, recent work introduced the *folding* strategy, which divides the dataset into several consecutive folds and applies sorting within each stage (Dai et al., 2025; Zhang et al., 2025). Following Dai et al. (2025), we tested the three-fold configuration they identified as optimal. These experiments are conducted with a 0.5B model. We replicate their observation that folding improves performance under a low peak LR ($1 \times 10^{-4}$) but find that this benefit diminishes—and even reverses—under a higher peak LR ($3 \times 10^{-3}$), as shown in Table 12. The higher-LR setting shows better overall performance than the low-LR setting in our experiments. These findings, as well as results of 1.5B models in Section 2, suggest that folding is not robust across different model scales or learning-rate settings. Its apparent gains may reflect compensation for a suboptimal LR schedule rather than

Table 12: Effect of the *folding* strategy under different peak learning rates. The benefit observed at a low LR ($1 \times 10^{-4}$) diminishes or reverses at a higher LR ($3 \times 10^{-3}$), indicating the limited robustness of folding across scales.

| Order | Strategy | Peak LR | MMLU | ARC-c | ARC-e | CSQA | OBQA | PIQA | SIQA | Wino. | Avg. |
|-------|----------|---------|------|-------|-------|------|------|------|------|-------|------|
| Uniform | – | $1 \times 10^{-4}$ | 25.70 | 28.43 | 37.72 | 34.32 | 30.20 | 61.81 | 40.99 | 50.83 | 38.75 |
| Ascend | Sorting | $1 \times 10^{-4}$ | 26.57 | 28.76 | 38.42 | 35.63 | 28.80 | 61.97 | 41.40 | 50.36 | 38.99 |
| Ascend | Folding | $1 \times 10^{-4}$ | 25.69 | 29.43 | 38.77 | 35.22 | 32.20 | 61.43 | 40.99 | 50.04 | 39.22 |
| Uniform | – | $3 \times 10^{-3}$ | 28.68 | 33.78 | 50.35 | 45.95 | 36.60 | 68.66 | 43.65 | 53.35 | 45.13 |
| Ascend | Sorting | $3 \times 10^{-3}$ | 27.78 | 37.12 | 47.89 | 44.47 | 37.40 | 67.85 | 43.30 | 55.80 | 45.20 |
| Ascend | Folding | $3 \times 10^{-3}$ | 28.33 | 33.11 | 48.25 | 43.82 | 38.80 | 69.21 | 43.76 | 52.88 | 44.77 |

a fundamental advantage. The limitations of the folding strategy and the strengths of our design underscore the importance of jointly considering curriculum design and LR scheduling.

**Interpretation via the Loss Landscape.** Our interpretation aligns with the river-valley model of the loss landscape (Wen et al., 2025), which characterizes optimization as progress along two primary directions: the *signal* (river) direction, where loss decreases gradually, and the *noise* (valley) direction, where loss oscillates sharply. We extend this framework by positing that data quality influences the gradient's components: high-quality data are assumed to provide a stronger, more stable signal direction and less noise, whereas low-quality data provide a weaker signal and induce more noise. In the context of a data curriculum, as data quality increases, the update direction becomes more dominated by the stable signal component. This mechanism facilitates faster convergence, as observed in our experiments (Figures 1(a) and 8(a)). In the river-valley model, when LR decay is applied, the optimizer settles toward the valley floor, reducing noise but also slowing progress along the signal direction, thus underusing the signal from high-quality data. Dremov et al. (2025) also analyzes the bias-variance relation during the cooldown phase in WSD schedule pretraining, using the river-valley framework.

## F    PROOFS IN SECTION 4

In Section 4, we analyze the bounds of expected loss under four different optimization cases:

1. **Uniform Sampling + Learning Rate Schedule**.
2. **Ascending Data-Ordering + Practical WSD Schedule**.
3. **Ascending Data-Ordering + WSMD Schedule**.
4. **Ascending Data-Ordering + Stochastic Weight Averaging (SWA)**.

In the following, we give the proof of their corresponding theoretical claims we mentioned in Section 4 one by one.

**Lemma F.1.** *Consider the uniform sampling, for any learning rate schedule $E$ such that $0 \le \eta_i \le 1$, and the parameter initialized at $(L, 0)$, it holds that*

$$\min_E \bar{\mathcal{L}}(M; E) = \Omega(L^2).$$

*Proof.* We first consider the update rule of SGD in the optimization process on the x-axis as

$$w_t^{(1)} = w_{t-1}^{(1)} - \eta_t(w_{t-1}^{(1)} - x_t^{(1)}).$$

Then, taking the expectation over the randomness in SGD and the data generation gives

$$\mathbb{E}[w_t^{(1)}] = (1 - \eta_t)\mathbb{E}[w_{t-1}^{(1)}] + \eta_t\mathbb{E}[x_t^{(1)}]$$

$$= (1 - \eta_t)\mathbb{E}[w_{t-1}^{(1)}] + \eta_t\frac{(M-1)d}{2}$$

$$\ge \frac{(M-1)d}{2}.$$

The last inequality holds because $\mathbb{E}[w_{t-1}^{(1)}] \ge \frac{(M-1)d}{2}$ can be shown by induction. Thus, we write out the lower bound for the expected loss

$$\mathbb{E}[\mathcal{L}(\boldsymbol{w}_t)] = \mathbb{E}[\|\boldsymbol{w}_t\|_2^2] \ge \mathbb{E}[(w_t^{(1)})^2] \ge (\mathbb{E}[w_t^{(1)}])^2 = \Theta(L^2).$$

The above equation completes the proof of Equation (1).    □

For Case 2 and Case 3, we give a more general lemma, for which the conclusions for Case 2 and Case 3 are direct corollaries.

**Lemma F.2.** *Consider the Ascending data-ordering, and a class of WSD learning rate schedules with the following formula*

$$\eta_t = \begin{cases} \eta_0 & \text{for } 1 \le t \le M - T_0 + 1 \\ \frac{1}{t - (M - T_0)} & \text{for } M - T_0 + 2 \le t \le M, \end{cases}$$

*where $T_0 = \omega(1)$, $M - T_0 = \Theta(M)$ and $\eta_0 = \frac{1}{2}$, it holds for any learning rate schedule $E$ with the above formula that*

$$\bar{\mathcal{L}}(M; E) = \tilde{\Theta}\left(T_0^2 d^2 + \frac{L^2}{T_0}\right).$$

*Proof.* We write out the update rule on the x-axis in the Ascending data-ordering case

$$w_t^{(1)} = (1 - \eta_t)w_{t-1}^{(1)} + \eta_t x_1^{(M-t+1)}.$$

Using the above update rule, we can get the expression of $w_M^{(1)}$ as

$$w_M^{(1)} = \prod_{t=M-T_0+2}^{M} (1 - \eta_t)w_{M-T_0+1}^{(1)} + \sum_{i=M-T_0+2}^{M} \prod_{j=i+1}^{M} (1 - \eta_j)\eta_i x_1^{(M-i+1)}.$$

Then, plugging in the formula of the learning rate schedule gives

$$
\begin{aligned}
w_M^{(1)} &= \prod_{t=M-T_0+2}^{M} \left(1 - \frac{1}{t-(M-T_0)}\right) w_{M-T_0+1}^{(1)} + \sum_{i=M-T_0+2}^{M} \prod_{j=i+1}^{M} (1-\eta_j)\eta_i x_1^{(M-i+1)} \\
&= \prod_{k=2}^{T_0} \frac{k-1}{k} w_{M-T_0+1}^{(1)} + \sum_{i=M-T_0+2}^{M} \frac{1}{T_0} x_1^{(M-i+1)} \\
&= \frac{1}{T_0} w_{M-T_0+1}^{(1)} + \sum_{i=M-T_0+2}^{M} \frac{1}{T_0} x_1^{(M-i+1)}.
\end{aligned}
$$

The above equation uses the following fact

$$
\begin{aligned}
&(1 - \frac{1}{T_0}) \cdot (1 - \frac{1}{T_0-1}) \cdots (1 - \frac{1}{i+1}) \cdot \frac{1}{i} \\
&= \frac{T_0-1}{T_0} \cdot \frac{T_0-2}{T_0-1} \cdots \frac{i}{i+1} \cdot \frac{1}{i} = \frac{1}{T_0},
\end{aligned}
$$

where $2 \leq i \leq T_0$. Furthermore, we can find that

$$
\begin{aligned}
w_{M-T_0+1}^{(1)} &= \prod_{t=1}^{M-T_0+1} (1-\eta_t) w_0^{(1)} + \sum_{i=1}^{M-T_0+1} \prod_{j=i+1}^{M-T_0+1} (1-\eta_j)\eta_i x_1^{(M-i+1)} \\
&= \frac{1}{2^{M-T_0+1}} w_0^{(1)} + \frac{1}{T_0} \sum_{i=1}^{M-T_0+1} \frac{1}{2^{M-T_0-i+1}} x_1^{(M-i+1)},
\end{aligned}
$$

and then

$$
\begin{aligned}
w_M^{(1)} &= \frac{1}{T_0} \sum_{i=M-T_0+2}^{M} x_1^{(M-i+1)} + \frac{1}{T_0} w_{M-T_0+1}^{(1)} \\
&= \frac{1}{T_0} \sum_{i=M-T_0+2}^{M} x_1^{(M-i+1)} + \frac{1}{T_0} \left( \frac{1}{2^{M-T_0+1}} w_0^{(1)} + \frac{1}{T_0} \sum_{i=1}^{M-T_0+1} \frac{1}{2^{M-T_0-i+1}} x_1^{(M-i+1)} \right) \\
&= \frac{1}{T_0} \frac{1}{2^{M-T_0+1}} w_0^{(1)} + \frac{1}{T_0} \sum_{i=1}^{T_0-1} x_1^{(i)} + \frac{1}{T_0^2} \sum_{i=T_0}^{M} \frac{1}{2^{i-T_0}} x_1^{(i)}.
\end{aligned}
$$

Given that $x_1^{(i)} = (i-1)d$, and $M - T_0 = \Theta(M), T_0 = \omega(1)$, we can find that

$$
\begin{aligned}
w_M^{(1)} &= \frac{1}{T_0} \frac{Md}{2^{M-T_0+1}} + \frac{1}{T_0} \sum_{i=1}^{T_0-1} (i-1)d + \frac{1}{T_0^2} \sum_{i=T_0}^{M} \frac{1}{2^{i-T_0}} (i-1)d \\
&= \frac{Md}{T_0 2^{M-T_0+1}} + \frac{(T_0-1)(T_0-2)d}{2T_0} + \frac{1}{T_0^2} \sum_{i=T_0}^{M} \frac{1}{2^{i-T_0}} (i-1)d.
\end{aligned}
$$

Now we analyze each term:

- First term: $\frac{Md}{T_0 2^{M-T_0+1}} = o(d)$ since $M - T_0 = \Theta(M)$ and $2^{M-T_0}$ grows exponentially.

- Second term: $\frac{(T_0-1)(T_0-2)d}{2T_0} = \Theta(T_0 d)$.

- Third term: Let $j = i - T_0$, then

$$\frac{1}{T_0^2} \sum_{i=T_0}^{M} \frac{1}{2^{i-T_0}} (i-1)d = \frac{d}{T_0^2} \sum_{j=0}^{M-T_0} \frac{1}{2^j} (T_0 + j - 1)$$

$$= \frac{d}{T_0^2} \left[ (T_0 - 1) \sum_{j=0}^{M-T_0} \frac{1}{2^j} + \sum_{j=0}^{M-T_0} \frac{j}{2^j} \right]$$

$$= \frac{d}{T_0^2} \left[ 2(T_0 - 1) + 2 + o(1) \right] = \frac{2T_0 d}{T_0^2} + o\left(\frac{d}{T_0}\right) = \Theta\left(\frac{d}{T_0}\right).$$

Therefore, $w_M^{(1)} = \Theta(T_0 d) + \Theta\left(\frac{d}{T_0}\right) = \Theta(T_0 d)$ since $T_0 = \omega(1)$.

Thus, the expected loss on the x-axis follows

$$\mathbb{E}[(w_M^{(1)})^2] = \Theta((T_0 d)^2) = \Theta(T_0^2 d^2).$$

Similarly, we write out the expected loss on the y-axis. Note that for $w_M^{(2)}$, the update rule is:

$$w_M^{(2)} = \frac{1}{T_0} \sum_{i=1}^{T_0-1} x_2^{(i)} + \frac{1}{T_0^2} \sum_{i=T_0}^{M} \frac{1}{2^{i-T_0}} x_2^{(i)} + o(d)$$

Since $x_2^{(i)} \sim \text{Uniform}(-L, L)$ and are independent, we have:

$$\mathbb{E}[(w_M^{(2)})^2] = \frac{1}{T_0^2} \sum_{i=1}^{T_0-1} \mathbb{E}[(x_2^{(i)})^2] + \frac{1}{T_0^4} \sum_{i=T_0}^{M} \frac{1}{2^{2(i-T_0)}} \mathbb{E}[(x_2^{(i)})^2]$$

$$= \frac{1}{T_0^2} \cdot (T_0 - 1) \cdot \frac{L^2}{3} + O\left(\frac{1}{T_0^4}\right) = \Theta\left(\frac{L^2}{T_0}\right).$$

The above equation completes the proof. Specifically, taking $T_0 = M - \lfloor 0.9M \rfloor$ and $T_0 = \Theta(M^{\frac{2}{3}})$ gives the results in Equation (2) and Equation (3). □

In the end, we show how a simple SWA method can beat the practical WSD schedule, which is stated in Theorem 4.1

*Proof of Theorem 4.1.* We consider the constant learning rate $\eta_0 \leq 1$ and ascending data-ordering. The SGD update is:

$$\boldsymbol{w}_t = (1 - \eta_0)\boldsymbol{w}_{t-1} + \eta_0 \boldsymbol{x}^{(M-t+1)}.$$

The solution to this recurrence is:

$$\boldsymbol{w}_t = (1 - \eta_0)^t \boldsymbol{w}_0 + \eta_0 \sum_{i=1}^{t} (1 - \eta_0)^{t-i} \boldsymbol{x}^{(M-i+1)}.$$

We consider the average of the last $n$ weights:

$$\bar{\boldsymbol{w}}_M = \frac{1}{n} \sum_{k=0}^{n-1} \boldsymbol{w}_{M-k}.$$

Substituting the expression for $\boldsymbol{w}_{M-k}$:

$$\bar{\boldsymbol{w}}_M = \frac{1}{n} \sum_{k=0}^{n-1} \left[ (1 - \eta_0)^{M-k} \boldsymbol{w}_0 + \eta_0 \sum_{i=1}^{M-k} (1 - \eta_0)^{M-k-i} \boldsymbol{x}^{(M-i+1)} \right]$$

$$= \frac{1}{n} \sum_{k=0}^{n-1} (1 - \eta_0)^{M-k} \boldsymbol{w}_0 + \frac{\eta_0}{n} \sum_{k=0}^{n-1} \sum_{i=1}^{M-k} (1 - \eta_0)^{M-k-i} \boldsymbol{x}^{(M-i+1)}.$$

Changing the order of summation in the second term:

$$\sum_{k=0}^{n-1}\sum_{i=1}^{M-k}(1-\eta_0)^{M-k-i}\boldsymbol{x}^{(M-i+1)} = \sum_{i=1}^{M}\left(\sum_{k=0}^{\min(n-1,M-i)}(1-\eta_0)^{M-k-i}\right)\boldsymbol{x}^{(M-i+1)}.$$

For $i \le M - n$, the inner sum is:

$$\sum_{k=0}^{n-1}(1-\eta_0)^{M-k-i} = (1-\eta_0)^{M-i}\cdot\frac{1-(1-\eta_0)^n}{\eta_0}.$$

For $i > M - n$, the inner sum is:

$$\sum_{k=0}^{M-i}(1-\eta_0)^{M-k-i} = (1-\eta_0)^{M-i}\cdot\frac{1-(1-\eta_0)^{M-i+1}}{\eta_0}.$$

Thus, we have:

$$\bar{\boldsymbol{w}}_M = \frac{1}{n}\sum_{k=0}^{n-1}(1-\eta_0)^{M-k}\boldsymbol{w}_0$$

$$+ \frac{1}{n}\sum_{i=1}^{M-n}(1-\eta_0)^{M-i}[1-(1-\eta_0)^n]\boldsymbol{x}^{(M-i+1)}$$

$$+ \frac{1}{n}\sum_{i=M-n+1}^{M}(1-\eta_0)^{M-i}[1-(1-\eta_0)^{M-i+1}]\boldsymbol{x}^{(M-i+1)}.$$

Now we analyze the x-component $\bar{w}_M^{(1)}$. Note that $x_1^{(i)} = (i-1)d$ and $d = L/M$. The first term is negligible since $(1-\eta_0)^{M-k}$ decays exponentially. For the second term, when $i \le M - n$, we have $(1-\eta_0)^{M-i} \le (1-\eta_0)^n$. Since $n = \Theta(M^{2/3})$, this term is exponentially small. The main contribution comes from the third term where $i > M - n$, i.e., the last $n$ data points. In this range, $(1-\eta_0)^{M-i}$ is not small, and we have:

$$\bar{w}_M^{(1)} \approx \frac{1}{n}\sum_{i=M-n+1}^{M}[1-(1-\eta_0)^{M-i+1}]x_1^{(M-i+1)}$$

$$= \frac{1}{n}\sum_{j=1}^{n}[1-(1-\eta_0)^j]x_1^{(j)}$$

$$\le \frac{1}{n}\sum_{j=1}^{n}x_1^{(j)} = \frac{1}{n}\sum_{j=1}^{n}(j-1)d = \frac{(n-1)n}{2n}d = \Theta(nd) = \Theta\left(\frac{nL}{M}\right).$$

Since $n = \Theta(M^{2/3})$, we have $\bar{w}_M^{(1)} = \Theta(L/M^{1/3})$, so:

$$\mathbb{E}[(\bar{w}_M^{(1)})^2] = \Theta\left(\frac{L^2}{M^{2/3}}\right).$$

For the y-component $\bar{w}_M^{(2)}$, note that $x_2^{(i)} \sim \text{Uniform}(-L, L)$ are independent. The variance of $\bar{w}_M^{(2)}$ is:

$$\text{Var}(\bar{w}_M^{(2)}) = \frac{1}{n^2}\sum_{j=1}^{n}[1-(1-\eta_0)^j]^2\text{Var}(x_2^{(j)})$$

$$\le \frac{1}{n^2}\sum_{j=1}^{n}\text{Var}(x_2^{(j)}) = \frac{1}{n^2}\cdot n\cdot\frac{L^2}{3} = \Theta\left(\frac{L^2}{n}\right) = \Theta\left(\frac{L^2}{M^{2/3}}\right).$$

Therefore, the total expected loss is:

$$\mathbb{E}[\mathcal{L}(\bar{\boldsymbol{w}}_M)] = \mathbb{E}[(\bar{w}_M^{(1)})^2] + \mathbb{E}[(\bar{w}_M^{(2)})^2] = \tilde{O}\left(M^{-\frac{2}{3}}L^2\right).$$

This completes the proof. $\qquad\square$

