# OpenReview forum: "How Learning Rate Decay Wastes Your Best Data in Curriculum-Based LLM Pretraining"
_ICLR.cc/2026/Conference — ICLR 2026 Oral_

### Official Review · Reviewer_QrZN · 2025-10-19

**Soundness:** 3
**Presentation:** 3
**Contribution:** 3
**Rating:** 6
**Confidence:** 4

**Summary:**

This paper argues that **curriculum-based pretraining** (train low→high quality) quietly clashes with the usual **decaying learning-rate (LR)** schedules: when the best data finally arrives (late in training), the LR is tiny, so the model barely learns from it. Experiments with a **1.5B** model on **30B** tokens show that curricula look strong under a **constant LR**, but their gains **shrink under cosine/WSD decay** (Figure 1–2). The authors propose **Curriculum Model Averaging (CMA)**: keep a high/constant LR during training, then do **model averaging** over the last checkpoints (SMA/EMA/WMA) to stabilize. They further recommend **co-designing** a *moderate* LR decay with model averaging (**CDMA**), which outperforms standard decay with uniform order and also beats “decay + curriculum” alone, with especially clear gains in **mid-training** when only some high-quality data is available (Table 1–2; Figure 5). A simple theory sketch supports why curriculum + averaging can keep strong updates from high-quality data while still reducing noise.

**Strengths:**

- The paper **clearly diagnoses** an intuitive but under-discussed coupling: high-quality data arrives when LR is tiny, dulling its impact. The **constant-LR** comparisons make this visible.
- **CMA** is **simple and actionable**: keep LR high, then **EMA/SMA** the last checkpoints; implementation details are explicit.
- **CDMA** (moderate decay + averaging) finds an **under-explored optimum** that beats standard decay+uniform and decay+curriculum, especially in **mid-training**.

**Weaknesses:**

- The study focuses on **one model size (1.5B) and a 30B-token corpus**; it’s unclear whether the same sweet spots hold for much larger models or different corpora/metrics.
- **Baselines are limited** for some comparisons (e.g., stronger **learned curricula** or **adaptive/variance-aware** data-selection methods aren’t included), so it’s hard to judge competitiveness against the latest dynamic strategies.
- The **mid-training** setting is promising but still tied to the specific phasing and quality signals here; more domains or public corpora would help establish external validity.

**Questions:**

Do the CMA/CDMA gains persist for **larger models** (e.g., 7B/13B) and for other **quality metrics** (beyond DCLM/PreSelect)? Please include at least one bigger-model run.

---

> ### Author Response · Authors · 2025-11-26
> **Response to Reviewer QrZN**
>
> We greatly thank the reviewer for appreciating our diagnosis of the problem and the simple and actionable strategies we propose. Your suggestions and feedback are important for us to clarify our contributions and make them convincing.
>
> **Q1: Does CMA/CDMA hold for larger models/scale?**
>
> **A1: Thanks for your suggestion, we perform our strategy on a close-to-real-world training setup, at a scale of a 3B model over 150B+ data tokens. The experiment scale and setup strongly support the results for our conclusion.** The details of our experiments are in `General Response`.
>
> **Q2: Can the method be generalized to other domains or quality metrics?**
>
> **A2: In the experiment in `General Response`, we incorporate a public Chinese data corpus (Fineweb-Edu-Chinese-V2.1) as well as their scoring metrics (fineweb score). We find improvement in the Chinese ability benchmark (CEval) for our strategy (a multi-domain variant) over the baseline.** We believe the result on a larger scale, on a quite different domain of dataset with a different quality signal, is convincing evidence for our approaches.
>
> **Q3: Baselines are limited, not including learned curricula or adaptive/variance-aware data selection.**
>
> **A3: We appreciate the reviewer's insightful comment regarding baseline comparisons. Our work aims to uncover and reconcile a fundamental conflict between LR decay and data curriculum, rather than proposing new curriculum designs.**
>
> To directly address the concern about baselines, we emphasize that our experiments already include recent state-of-the-art curriculum learning approaches, specifically the folding/interleaving curriculum methods from [1,2]. Our analysis reveals that these advanced methods still face the same fundamental conflict between data curriculum and learning rate decay that we identify in our work. The benefits of folding curriculum in certain circumstances stem from a trade-off between curriculum advantages and LR decay effects, as detailed in the last paragraph of Section 2 and Appendix E.1 of our revised paper.
>
> In addition, we notice that in these recent works on curriculum design, they primarily focus on score design or offline curriculum strategies. We conjecture that adaptive curriculum methods still face challenges in scaling up. Existing adaptive curricula often rely on online selection, which will introduce additional overhead and face challenges to scale up in practical pretraining.
>
>
> Nevertheless, we would also be interested if the reviewer could specify some practically used and high-performing adaptive curriculum strategies for our future exploration. Actually, our findings provide insights that can directly benefit future adaptive curriculum and learned data selection designs. Specifically, our work highlights that LR scheduling and data ordering should be coordinated rather than optimized independently for synergistic benefits, suggesting that future adaptive curricula could incorporate LR–data co-adaptation for improved stability and generalization.
>
> Therefore, our chosen baselines, including uniform ordering with LR decay, curriculum with LR decay, and uniform ordering with model averaging, appropriately reflect primary and practically used settings for analyzing training dynamics. While learned curricula represent an interesting direction, their inclusion would not change our fundamental findings about the LR-curriculum interplay.
>
> [1] [Beyond Random Sampling: Efficient Language Model Pretraining via Curriculum Learning](https://arxiv.org/abs/2506.11300)
>
> [2] [Data Efficacy for Language Model Training](https://arxiv.org/abs/2506.21545)

---

### Official Review · Reviewer_qZNE · 2025-10-20

**Soundness:** 3
**Presentation:** 3
**Contribution:** 3
**Rating:** 6
**Confidence:** 4

**Summary:**

This paper investigates a critical and often-overlooked issue in curriculum-based LLM pretraining: the detrimental coupling between the data curriculum and the learning rate (LR) schedule. The authors diagnose that standard LR decay schedules, which are designed to reduce noise and stabilize training, conflict with the goal of curriculum learning (CL), which places the most valuable, high-quality data at the end of training. They frame this conflict through the lens of the LR's "dual role," acting simultaneously as an update step size and an implicit importance weight for data. By decaying the LR, standard methods effectively "waste" the best data by learning from it with minimal update steps.

To resolve this, the paper proposes to decouple these two roles. Their primary solution, Curriculum Model Averaging (CMA), replaces aggressive LR decay with a constant high LR during training, relying on model averaging (e.g., EMA or SMA) over the final checkpoints to ensure stability. They further propose a co-designed strategy, CDMA, which combines a moderate LR decay with model averaging. Through extensive experiments on a 1.5B parameter LLM, the authors demonstrate that their proposed methods outperform standard pretraining baselines, especially when compared to the widely-used cosine schedule. The work identifies an "optimal area" of moderate decay where co-designing the curriculum, LR schedule, and model averaging yields the best results, highlighting a previously underexplored optimization regime.

**Strengths:**

1.  The paper's primary strength lies in its clear and intuitive diagnosis of a fundamental conflict in modern pretraining.

2.  The authors provide strong empirical validation for their central thesis.

3.  The proposed solutions, CMA and CDMA, are not ad-hoc but are direct, logical consequences of the initial diagnosis.

**Weaknesses:**

1.  While the paper's framing is compelling, the core ideas—the conflict between CL and LR decay, and the use of model averaging for stabilization—are not entirely new. The provided literature survey highlights several precedents. For instance, Weinshall & Amir (2020) theoretically showed that optimal CL requires non-decaying LRs, and Jiang et al. (2021) empirically demonstrated that LR decay undervalues late-stage data. Similarly, model averaging techniques like SWA (Izmailov et al., 2018) have long been proposed as alternatives to LR decay. The paper could strengthen its contribution by more explicitly positioning its work against these specific precedents.

2.  The experiments are missing a key and highly relevant baseline: training exclusively on a high-quality data subset for the same computational budget (e.g., by filtering out the bottom 80% of data and repeating the top 20%). Without this comparison, it is difficult to disentangle the benefits of the curriculum's data ordering from the benefits of simply focusing compute on high-quality data. If this simple filtering baseline performs comparably to CMA/CDMA, it would challenge the necessity of the curriculum itself.

3.  The paper observes that a descending (high-to-low quality) curriculum performs poorly but misses an opportunity for deeper analysis. This ordering, when paired with a standard LR decay, seems intuitively synergistic (high LR on high-quality data, low LR on low-quality data).

---

[1] Weinshall, D., & Amir, G. (2020). On the Theory of Curriculum Learning. *Advances in Neural Information Processing Systems*.

[2] Jiang, L., et al. (2021). Prioritized Training on Points that are Learnable, Worth Learning, and Not Yet Learnt. *International Conference on Machine Learning*.

[3] Izmailov, P., et al. (2018). Averaging Weights Leads to Wider Optima and Better Generalization. *Uncertainty in Artificial Intelligence*.

**Questions:**

see weakness

---

> ### Author Response · Authors · 2025-11-26
> **Response to Reviewer qZNE [1/3]**
>
> We would like to thank the reviewer for their appreciation of the key novelty of our work and recognition of our extensive empirical validation. Moreover, your valuable and constructive suggestions are very important for us to further improve our work.
>
> **Q1: Strengthen the contribution by more explicitly positioning the work against some specific precedents listed above.**
>
> **Question over References from Reviewer in Q1: We find some flaws in the references and claims in Weakness 1, and we hope the reviewer `qZNE` will clarify.**
>
> 1. We did not find the explicit claims in [1] that *theoretically showed that optimal CL requires non-decaying LRs* and in [2] that *demonstrated that LR decay undervalues late-stage data*. It would be kind of you to make your concerns more specific for these references.
>
> 2. The title of reference [1] does not exist. According to the information you provided, we find the closest reference is
>
>     [1] Weinshall, D., & Amir, G. (2020). Theory of Curriculum Learning, with Convex Loss Functions. *Journal of Machine Learning Research*.
>
>     We did not find its version in *Advances in Neural Information Processing Systems*.
>
> 3. The year and authors of reference [2] you provided seem mistaken. According to the title, we find no authors in [2] named Jiang, L., and the publication year should be 2022.
>
>
> **Nevertheless, we have added discussion about precedents as mentioned in `Summary of Revisions` and provided discussion for the specific precedents you mentioned as follows.**
>
> **A1: Our approach distinguishes itself from precedents in (1) identifying the conflict between CL and LR decay with extensive empirical validation; (2) combining weight averaging and curriculum learning and finding synergistic benefits from this combination; (3) compatibility with the LLM pretraining pipeline, taking advantage of the data preprocessing pipeline.** We discuss in detail as follows.
> 1. Weinshall & Amir[1] primarily propose the concept of Ideal Difficulty Score and Local Difficulty, and analyze the correlation of convergence rate and difficulty score. However, their results are derived from linear regression and classification by hinge loss minimization settings. They do not underscore the conflict between data quality/difficulty and the LR decay schedule. In comparison, we emphasize the conflict between the LR schedule and the data curriculum, and conduct extensive experiments over modern LLM architecture to validate our results, which are much closer to a practical and modern training setting.
> 2. Mindermann et al.[2] propose an online batch selection approach, using reducible holdout loss (RHO-LOSS), which computes the gap between training loss and irreducible holdout loss as a metric for data selection. Our approach is somewhat orthogonal to RHO-LOSS, because we focus on reconciling the conflict between training dynamics (affected by LR decay) and data distribution (determined by data curriculum), instead of searching for a better online/offline data metric. Moreover, our approach is compatible with the prevailing pretraining paradigm, taking advantage of the quality information from the data preprocessing phase, while RHO-LOSS requires a modification of the training pipeline. Therefore, our approach is in plug-and-play manner and more efficient. Finally, inspired by your suggestions, we believe it can be a promising direction to design adaptive LR schedules and online batch selection collaboratively to better align training dynamics and data distribution.
> 3. SWA[3] proposes the weight averaging strategy and discovers better generalization ability obtained from wider optima. As discussed in Appendix B.3 in our submission version, our work dives into the interaction between model averaging and data quality, which is largely ignored by previous works. Moreover, this combination is not a naive A+B addition, as discussed in detail in Section 5.1, Paragraph 2 in our submission version, the model averaging and data quality can exhibit synergistic benefits, while only one component cannot take effect extensively. Therefore, our contribution to the model averaging technique lies in the incorporation of a curriculum learning strategy, to amplify the benefit from model averaging, especially for the modern LLM pretraining setup.
>
> [1] [Theory of Curriculum Learning, with Convex Loss Functions](https://arxiv.org/abs/1812.03472)
>
> [2] [Prioritized Training on Points that are Learnable, Worth Learning, and Not Yet Learnt](https://arxiv.org/abs/2206.07137)
>
> [3] [Averaging Weights Leads to Wider Optima and Better Generalization](https://arxiv.org/abs/1803.05407)

---

> ### Author Response · Authors · 2025-11-26
> **Response to Reviewer qZNE [2/3]**
>
> **Q2: Consider a baseline that repeats more-aggressively filtered data and keeps the same computational budget.**
>
> **A2: The repeating high-quality data baseline is orthogonal to our approach because we primarily consider pretraining over filtered data after preprocessing. Nevertheless, we conduct experiments for your proposed setup and find that our strategy can outperform the baseline on both Core and Avg benchmark scores.**
>
> Our method primarily assumes the data has gone through the data preprocessing pipeline and focuses on the training strategy over the filtered data. If mixing the multi-epoch training, curriculum learning, and data filtering all at once, it would be too complex to understand them clearly. Our proposed approach is to use curriculum learning and model averaging to improve the training efficiency, under the assumption that the data filtering has been completed. Thus, the multi-epoch training strategy can be an orthogonal strategy regarding our approach, and the combination of our approach and multi-epoch training can be an interesting direction for future exploration.
>
> In this sense, we can actually discuss your concerns and interpret them as the following question: *In our setting, can we achieve better performance through more aggressive data filtering combined with multi-epoch pretraining to achieve a better result?*
>
>
>
> Although this question does not strongly correlate with our work, we are also interested in the results and conducted experiments to share with you here. We keep top-score samples in the DCLM subset according to a specific quality score and a retention ratio, and repeat these data in multiple epochs to keep the total training FLOPs the same. The data samples are in uniform order within the dataset. We call this strategy `Filter&Repeat`. We ablate over the quality metrics (DCLM score and PreSelect score) and the filtering ratios over the original dataset. We find that the multi-epoch strategy on aggressively filtered data can actually generate better results on some benchmarks. For example, keeping the top 13.8% and repeating can result in better MMLU. However, this method can result in performance drops in other benchmarks (possibly due to overfitting) and lead to degradation on average. We sweep some retaining ratios and find that the best combination still cannot match our proposed result, which goes through one pass over the dataset. The evaluation benchmark table is as follows.
>
>
> | Method | Quality Score | Retain Ratio | MMLU | ARC-c | ARC-e | CSQA | OBQA | PIQA | SIQA | Wino. | Avg. | Core |
> |---------------|---------------|---------------|------|---------------|----------|------|------------|------|------------|------------|------|------|
> | Filter&Repeat | DCLM          | 13.8\%         | **32.99** | 35.79 | 61.75 | 46.03 | 42.00 | 71.71 | 44.37 | 56.35 | 48.87 | 44.14 |
> | Filter&Repeat | PreSelect     | 18.6\%         | 32.24 | 37.79 | 61.23 | 46.03 | 42.40 | 71.76 | 44.32 | 55.80 | 48.95 | 44.32 |
> | Filter&Repeat | DCLM          | 33.4\%         | 32.44 | 41.14 | **61.93** | 51.11 | 43.80 | 72.09 | 45.34 | **58.80** | 50.83 | 46.65 |
> | Filter&Repeat | DCLM          | 77.4\%         | 31.68 | 38.46 | 60.70 | **52.50** | 45.00 | **72.52** | **45.80** | 57.22 | 50.49 | 45.83 |
> | CMA| DCLM  | 100% | 31.68 | **41.47** | **61.93** | **52.50** | **46.00** | 71.71 | 45.39 | 57.22 | **50.99** | **46.89** |

---

> ### Author Response · Authors · 2025-11-26
> **Response to Reviewer qZNE [3/3]**
>
> **Q3: Why does Descend+Decay not exhibit good performance? Especially consider that it matches the LR schedule and data curriculum.**
>
> **A3: There is a trade-off between forgetting and matching the LR schedule and data curriculum. In practice, the performance hurt from forgetting can outweigh the benefits from matching the LR schedule and data curriculum.** We conduct the following experiments to support our conclusions.
>
> (1) **The synergistic benefit you mention does hold for the descending curriculum and LR decay schedule.** Note that we always incorporate warmup steps in our training setting. Although the warmup phase only consists of a small ratio ($\le 0.05$), we adjust the data distribution for better matching schedules for descending curriculum training, putting the best data after the warmup phases and filling the previous ending data (low-quality part) into the warmup phase. In this sense, the warmup phase is filled with low-quality data, and the best data comes at the peak of the learning rate schedule. We adopt the Cosine LR schedule for these experiments. We find that in this way, **after better matching schedules, the descend curriculum can perform better than reported in the original version.** This observation further supports our conclusion on the relationship between LR decay and data curriculum. The skip-warmup descending can outperform previous descending **but still underperform uniform data ordering.** The result table is as follows:
>
> |    | MMLU | ARC-c | ARC-e | CSQA | OBQA | PIQA | SIQA | Wino. | Avg. | Core |
> |-----------------------|------|---------------|----------|------|------------|------|------------|------------|------|------|
> | skip-warmup descend   | 30.35 | 37.46 | 54.74 | **49.96** | **43.40** | 71.93 | 45.14 | **57.54** | 48.81 | 43.13 |
> | descend               | 29.51 | 34.11 | 52.98 | 48.81 | 42.60 | **72.42** | **45.45** | 55.17 | 47.63 | 41.36 |
> | uniform               | **30.49** | **38.13** | **59.47** | 49.14 | 42.20 | 71.87 | 45.19 | 56.51 | **49.13** | **44.31** |
>
> (2) **The forgetting problem overall matters in practice, and thus the descending curriculum can be hurt by the reverse ordering, even though the best data samples are introduced when LR is high.** We conduct experiments on a 0.5B model over 10B data, and run the following ablations by adjusting the descending curriculum in the following ways: we move the best data chunk (about 10% of the total) to the start, middle, and end of the training run. `start` means the best data occurs right after the warmup phase. `middle` means the best data is introduced in the middle of training, and `end` means putting the best data chunk at the training end. We adopt WSD decay schedules in these experiments, so `start` and `mid` introduce high-quality data at peak LR, but `end` introduces high-quality data at lower LR. We find that, as we put the best data closer to the end, the overall performance of the model can increase, while the LR schedule and relative ordering of other data samples are kept. Note that the performance rankings are not consistent across the benchmarks, which may reflect the different trade-offs between forgetting and matching the LR schedule and data curriculum for different benchmarks. In this sense, we can support our conclusions that forgetting can outweigh the benefit of matching the LR schedule and data curriculum in the `Descend+LR Decay` case. The result table is as follows:
>
> |  | MMLU | ARC-c | ARC-e | CSQA | OBQA | PIQA | SIQA | Wino. | Avg. | Core |
> |---------------|------|---------------|----------|------|------------|------|------------|------------|------|------|
> | start         | 26.34 | 24.75 | 44.04 | 42.51 | **40.20** | 67.95 | **43.50** | 51.93 | 42.65 | 34.41 |
> | mid           | 26.97 | 30.77 | 42.46 | **46.19** | 33.60 | **68.28** | 43.04 | **52.17** | 42.93 | 36.60 |
> | end           | **27.33** | **33.78** | **48.95** | 44.96 | 38.40 | 66.54 | 42.99 | 51.85 | **44.35** | **38.76** |

---

### Official Review · Reviewer_vPHj · 2025-10-30

**Soundness:** 3
**Presentation:** 3
**Contribution:** 3
**Rating:** 6
**Confidence:** 3

**Summary:**

This paper investigates the overlooked interaction between data curriculum strategies and learning rate decay in LLM pretraining. The authors argue that when high-quality data are placed at the end of a curriculum schedule, the model fails to fully learn from these data since the LR has already decayed. To address this issue, the paper proposes Curriculum Model Averaging (CMA), which combines a quality-based curriculum with model averaging, replacing LR decay with CMA to focus the final model on high-quality signals. The authors also proposed CDMA, which combines CMA with LR decay. Experiments on a 1.5B-parameter model trained on 30B tokens show that CMA/CDMA consistently improve both validation loss and downstream task performance, especially on Core benchmarks (max.+2.50%).

**Strengths:**

The paper identifies a realistic but previously underexplored issue -- the negative coupling between LR decay and quality-based curricula. The authors proposed a simple yet effective method, using checkpoint averaging to counteract LR over-decay, without additional retraining or architecture changes. The approach can be easily integrated into standard pipelines, and is computationally inexpensive. Theoretical modeling and gradient trajectory visualization illustrate how the proposed method alleviates the loss of learning signal caused by LR decay in curriculum learning.

**Weaknesses:**

Results are mostly at 1.5B model parameters and 30B tokens. It remains uncertain whether the same trends hold for much larger models (e.g., 7B–70B) or longer training runs. The paper also lacks exploration of hyperparameters for checkpoint averaging (decay factor, checkpointing interval, number of checkpoints) or justification for the selected hyperparameter values.

**Questions:**

1. How does CMA/CDMA perform on larger models (e.g., >=7B) or longer training (e.g. 100B+ tokens)? It would be better if you provide at least one larger-scale validation experiment to improve the generality of your claims.
2. How did you tune the hyperparameters or select their specific values?

---

> ### Author Response · Authors · 2025-11-26
> **Response to Reviewer vPHj [1/3]**
>
> We would like to thank the reviewer for appreciating our CMA approach and sincerely thank you for your effort and thoughtful feedback on our work. Your comments and suggestions have greatly contributed to improving the quality and clarity of our paper.
>
> **Q1: It is better to provide a larger-scale validation experiment to improve the generality of claims.**
>
> **A1:** **Our strategy outperforms the baseline, at a scale of 3B model, trained on over 150B tokens in a continual training setting, as shown in `General Response`.** The experiment responds to your concerns in the following aspects:
>
> (1) The model size is over 3B, a scale matching practical edge-end LLM size.
>
> (2) The data size in the continual phase is over 150B, and the shared base phase consists of about 729B tokens. This dataset scale is sufficient to validate our claim.
>
> (3) The benchmark scores on selected datasets outperform the baseline overall by 1.09%. In particular, domains sorted by respective domain quality metrics (Fineweb-C, DCLM datasets) exhibit stable and consistent improvements in corresponding capabilities.
>
> (4) We train our model on a data mixture with different domains, which is also closer to a frontier pretraining setup.
>
> **Q2: How did you tune the hyperparameters for checkpoint averaging (decay factor, checkpointing interval, number of checkpoints) or select their specific values?**
>
> **A2: Model averaging does not need additional training requirements, so it is relatively cheaper to tune checkpoint hyperparameters. For hyperparameter selection, we mainly follow previous work[1,2], and we find the hyperparameters suffice for our setting after slight tuning.** To answer your questions, we will discuss two perspectives:
>
> *(1) How do these hyperparameters affect the performance of our methods?*
>
> *(2) How to select specific values for the hyperparameters?*
>
> We will discuss these two subproblems respectively as follows.

---

> ### Author Response · Authors · 2025-11-26
> **Response to Reviewer vPHj [2/3]**
>
> **Q2.1: How do these hyperparameters affect the performance of our methods?**
>
> **A2.1: Our strategy is robust around our selected hyperparameters, especially for CDMA.** We ablate the hyperparameters on CMA/CDMA in our default setting. Our default practice averages over the last 6 checkpoints, with an interval of about 0.21B tokens, using EMA with $\alpha=0.2$, and the model is trained on a shard of DCLM Baseline dataset.
>
> For completeness, our ablation experiments are conducted over variants of CMA/CDMA, including ending LRs at $1\times 10^{-3},2\times 10^{-3}, 3\times 10^{-3}$ (same as peak LR). We denote these variants as **CDMA1**, **CDMA2**, **CMA** respectively. We focus on three dimensions of hyperparameters for model averaging: checkpoint intervals, number of checkpoints, and decay factors. The composition of 3 dimensions is denoted as ($\Delta$, #, $\alpha$), where $\Delta$ denotes the interval range (measured by steps, one batch is approximately 2M), \# denotes the number of checkpoints, and $\alpha$ denotes the decay factor. The base configuration adopted in our work is `(100, 6, 0.2)`. We discuss our ablation results on checkpoint intervals, decay factor, and checkpoint number one by one.

---

> > ### Author Response · Authors · 2025-11-26
> > **Hyperparameter Ablation [1/3]: Checkpoint Interval**
> >
> > **In practice, checkpoint interval is not a fully free hyperparameter to tune; it depends on the efficiency of checkpointing and storage budget.** 0.21B is our unit checkpoint interval and we decide this range according to efficiency and our storage budget. Nevertheless, we conduct ablation experiments on the checkpoint intervals as follows.
> > (1) We take 1 checkpoint from every 2 checkpoints, and average over these 3 checkpoints with an interval of 0.42B tokens;
> > (2) For CMA, we rerun the last part of training, using a denser checkpoint interval of 0.084B tokens, and check results for 6 checkpoints with an interval of 0.084B tokens.
> >
> > We find checkpoint interval is robust for CDMA, but not robust when #ckpts is less & using CMA. The result details are as follows.
> >
> >
> > ---
> >
> > ### **CMA Results**
> >
> > | Configuration | MMLU | ARC-c | ARC-e | CSQA | OBQA | PIQA | SIQA | Wino. | Avg. | Core |
> > | ------------- | ----- | ------------- | -------- | ----- | ---------- | ----- | ---------- | ---------- | ----- | ----- |
> > | ($\Delta$=50, #=3) | 31.82 | 40.13 | 60.88 | 50.94 | 45.60 | 72.03 | 44.98 | 56.20 | 50.32 | 45.94 |
> > | ($\Delta$=10, #=6) | 31.85 | 42.47 | 61.75 | 50.70 | 44.60 | 71.49 | 45.85 | 57.46 | 50.77 | 46.69 |
> > | Base ($\Delta$=25, #=6) | 32.17 | 40.80 | 61.75 | 53.07 | 44.80 | 71.55 | 45.85 | 57.62 | 50.95 | 46.95 |
> >
> >
> > ---
> >
> > ### **CDMA1 Results** [ending LR = $1\times 10^{-3}$]
> >
> > | Configuration | MMLU | ARC-c | ARC-e | CSQA | OBQA | PIQA | SIQA | Wino. | Avg. | Core |
> > | ------------- | ----- | ------------- | -------- | ----- | ---------- | ----- | ---------- | ---------- | ----- | ----- |
> > | ($\Delta$=50, #=3) | 31.72 | 40.80 | 62.11 | 52.42 | 46.00 | 72.14 | 45.09 | 57.54 | 50.98 | 46.76 |
> > | Base ($\Delta$=25, #=6) | 31.68 | 41.47 | 61.93 | 52.50 | 46.00 | 71.71 | 45.39 | 57.22 | 50.99 | 46.89 |
> >
> > ---
> >
> > ### **CDMA2 Results** [ending LR = $2\times 10^{-3}$]
> >
> > | Configuration | MMLU | ARC-c | ARC-e | CSQA | OBQA | PIQA | SIQA | Wino. | Avg. | Core |
> > | ------------- | ----- | ------------- | -------- | ----- | ---------- | ----- | ---------- | ---------- | ----- | ----- |
> > | ($\Delta$=50, #=3) | 31.66 | 42.47 | 62.11 | 51.92 | 44.40 | 71.65 | 45.96 | 57.30 | 50.94 | 47.04 |
> > | Base ($\Delta$=25, #=6) | 31.81 | 40.47 | 62.46 | 51.60 | 43.80 | 71.60 | 46.21 | 58.17 | 50.76 | 46.58 |
> >
> > ---

---

> > ### Author Response · Authors · 2025-11-26
> > **Hyperparameter Ablation [2/3]: Decay Factor**
> >
> > **Setting decay factor around $\alpha=0.2$ is stable, especially for CDMA.** For each case, we ablate with $\alpha=0.1, 0.3$ around $\alpha=0.2$ in our work, and find the results are relatively stable over this hyperparameter.
> >
> > ---
> >
> > ### **CMA Results**
> >
> > | Configuration | MMLU | ARC-c | ARC-e | CSQA | OBQA | PIQA | SIQA | Wino. | Avg. | Core |
> > | ------------- | ----- | ------------- | -------- | ----- | ---------- | ----- | ---------- | ---------- | ----- | ----- |
> > | $\alpha$=0.1 | 31.97 | 41.14 | 61.40 | 52.50 | 45.40 | 71.82 | 44.98 | 56.75 | 50.75 | 46.75 |
> > | $\alpha$=0.2 (Base) | 32.17 | 40.80 | 61.75 | 53.07 | 44.80 | 71.55 | 45.85 | 57.62 | 50.95 | 46.95 |
> > | $\alpha$=0.3 | 31.77 | 42.14 | 61.05 | 51.76 | 44.80 | 71.82 | 44.93 | 56.91 | 50.65 | 46.68 |
> >
> > ---
> >
> > ### **CDMA1 Results** [ending LR = $1\times 10^{-3}$]
> >
> > | Configuration | MMLU | ARC-c | ARC-e | CSQA | OBQA | PIQA | SIQA | Wino. | Avg. | Core |
> > | ------------- | ----- | ------------- | -------- | ----- | ---------- | ----- | ---------- | ---------- | ----- | ----- |
> > | $\alpha$=0.1 | 31.61 | 40.47 | 61.75 | 52.09 | 45.40 | 71.71 | 45.34 | 57.54 | 50.74 | 46.48 |
> > | $\alpha$=0.2 (Base) | 32.17 | 40.80 | 61.75 | 53.07 | 44.80 | 71.55 | 45.85 | 57.62 | 50.95 | 46.95 |
> > | $\alpha$=0.3 | 31.68 | 41.14 | 61.93 | 52.17 | 45.20 | 71.71 | 44.98 | 57.14 | 50.74 | 46.73 |
> >
> > ---
> >
> > ### **CDMA2 Results** [ending LR = $2\times 10^{-3}$]
> >
> > | Configuration | MMLU | ARC-c | ARC-e | CSQA | OBQA | PIQA | SIQA | Wino. | Avg. | Core |
> > | ------------- | ----- | ------------- | -------- | ----- | ---------- | ----- | ---------- | ---------- | ----- | ----- |
> > | $\alpha$=0.1 | 31.85 | 42.14 | 62.63 | 51.76 | 43.80 | 71.76 | 46.06 | 58.09 | 51.01 | 47.10 |
> > | $\alpha$=0.2 (Base) | 31.81 | 40.47 | 62.46 | 51.60 | 43.80 | 71.60 | 46.21 | 58.17 | 50.76 | 46.58 |
> > | $\alpha$=0.3 | 31.74 | 40.80 | 62.63 | 51.68 | 44.00 | 71.60 | 45.91 | 58.33 | 50.84 | 46.71 |
> >
> > ---

---

> > ### Author Response · Authors · 2025-11-26
> > **Hyperparameter Ablation [3/3]: Number of Checkpoints**
> >
> > **The checkpoint number can affect the performance of CMA, especially when checkpoint number is less or using CMA.**
> > For each variant, we ablate experiments with checkpoint number as 4, 8, 10. The ablation results are listed as follows.
> >
> > ---
> >
> > ### **CMA Results**
> >
> > | Configuration | MMLU | ARC-c | ARC-e | CSQA | OBQA | PIQA | SIQA | Wino. | Avg. | Core |
> > | ------------- | ----- | ------------- | -------- | ----- | ---------- | ----- | ---------- | ---------- | ----- | ----- |
> > | #=4 | 31.44 | 39.80 | 60.70 | 50.53 | 45.40 | 71.65 | 45.85 | 57.70 | 50.38 | 45.62 |
> > | #=6 (Base) | 32.17 | 40.80 | 61.75 | 53.07 | 44.80 | 71.55 | 45.85 | 57.62 | 50.95 | 46.95 |
> > | #=8 | 31.75 | 39.80 | 61.05 | 51.11 | 46.20 | 71.82 | 45.04 | 57.46 | 50.53 | 45.93 |
> > | #=12 | 31.85 | 40.13 | 60.88 | 51.27 | 45.60 | 71.44 | 44.93 | 56.59 | 50.34 | 46.03 |
> >
> > ---
> >
> > ### **CDMA1 Results** [ending LR = $1\times 10^{-3}$]
> >
> > | Configuration | MMLU | ARC-c | ARC-e | CSQA | OBQA | PIQA | SIQA | Wino. | Avg. | Core |
> > | ------------- | ----- | ------------- | -------- | ----- | ---------- | ----- | ---------- | ---------- | ----- | ----- |
> > | #=4 | 31.66 | 39.80 | 61.58 | 51.02 | 44.80 | 71.82 | 44.98 | 57.22 | 50.36 | 46.02 |
> > | #=6 (Base) | 31.68 | 41.47 | 61.93 | 52.50 | 46.00 | 71.71 | 45.39 | 57.22 | 50.99 | 46.89 |
> > | #=8 | 31.82 | 41.14 | 62.11 | 52.25 | 45.60 | 71.76 | 45.24 | 56.67 | 50.82 | 46.83 |
> > | #=12 | 31.84 | 41.14 | 62.46 | 52.33 | 45.20 | 71.98 | 45.29 | 56.91 | 50.89 | 46.94 |
> >
> > ---
> >
> > ### **CDMA2 Results** [ending LR = $2\times 10^{-3}$]
> >
> > | Configuration | MMLU | ARC-c | ARC-e | CSQA | OBQA | PIQA | SIQA | Wino. | Avg. | Core |
> > | ------------- | ----- | ------------- | -------- | ----- | ---------- | ----- | ---------- | ---------- | ----- | ----- |
> > | #=4 | 31.84 | 41.81 | 62.63 | 50.45 | 44.00 | 71.11 | 45.19 | 58.88 | 50.74 | 46.68 |
> > | #=6 (Base) | 31.81 | 40.47 | 62.46 | 51.60 | 43.80 | 71.60 | 46.21 | 58.17 | 50.76 | 46.58 |
> > | #=8 | 31.91 | 40.80 | 62.63 | 51.60 | 44.00 | 71.65 | 45.91 | 58.72 | 50.90 | 46.73 |
> > | #=12 | 31.87 | 41.47 | 62.46 | 51.52 | 43.40 | 71.87 | 46.11 | 58.72 | 50.93 | 46.83 |
> >
> > ---
> >
> > The relative sensitivity of checkpoint number in CMA can be explained as follows:
> >
> > (1) The checkpoint number cannot be too low, probably due to the requirement to reduce noise;
> >
> > (2) The degradation of CMA with more checkpoints can be attributed to the degradation of checkpoints beyond the range of the last 6 checkpoints, as revealed by the ARC scores of the last checkpoints in CMA. The ARC task scores for the last 8 checkpoints in CMA are listed as follows; we find the performance boosts after 14900 steps, which supports the checkpoint number being 6 for the CMA strategy. Further increasing the checkpoint number can risk taking some immature checkpoints, and thus hurt model average performance.
> >
> > ---
> >
> > | Step | ARC-Challenge | ARC-Easy |
> > | ---- | ------------- | -------- |
> > | 14700 | 33.11 | 55.44 |
> > | 14800 | 33.11 | 55.79 |
> > | 14900 | 34.11 | 56.84 |
> > | 15000 | 38.13 | 57.89 |
> > | 15100 | 36.79 | 57.02 |
> > | 15200 | 34.11 | 58.60 |
> > | 15300 | 34.11 | 58.95 |
> > | 15375 | 38.46 | 56.14 |
> >
> > ---

---

> ### Author Response · Authors · 2025-11-26
> **Response to Reviewer vPHj [3/3]**
>
> **Q2.2: How to select specific values for the hyperparameters?**
>
> **A2.2: Our approach mainly follows previous work[1,2]; we find that the hyperparameters suffice for our setting after slight tuning. We also provide the guidelines for selecting the number of checkpoints here.**
> - The checkpoint interval selection depends on our storage budget and training efficiency, and is set to about 0.2B in our setting.
> - The decay factor $\alpha$ for EMA refers to the ablation study in previous work[1], in which $\alpha=0.2$ and $\alpha=0.1$ are discussed. With slight tuning, we find $\alpha=0.2$ is good enough for our setting with EMA.
> - The checkpoint number should be tuned accordingly, since the checkpoint interval can be different across various setups. As reported in previous works[1,2], too few checkpoints can result in degraded performance, primarily due to insufficient variance reduction among the checkpoints. On the other hand, *we propose the following simple guideline for checkpoint selection: we can evaluate the last several (e.g., $\ge 8$) checkpoints over **a lightweight subset of benchmarks**, like the ARC benchmark results shown above.* Then we can select the checkpoints accordingly to avoid including immature checkpoints. Note that we suggest using only a lightweight subset of benchmarks to avoid the risk of cherry-picking. Combining these two aspects, we set the default checkpoint number to 6 in our experiments.
>
> [1] [Model Merging in Pre-training of Large Language Models](https://arxiv.org/abs/2505.12082)
>
> [2] [WSM: Decay-FreeLearningRateSchedule via Checkpoint Merging for LLM Pre-training](https://arxiv.org/pdf/2507.17634)

---

### Author Response · Authors · 2025-11-26
**Summary of Revisions**

We sincerely thank all reviewers for their valuable feedback and constructive suggestions. Based on your comments and to facilitate our discussion, we have polished the paper and list the revisions as follows (section references correspond to the revised version):
- **Abstract & Section 1**: We rephrased the abstract to summarize our contributions and advantages more clearly; we reorganized the introduction and added footnotes to clarify concepts.
- **Section 3.1**: We discuss the role of moderate LR decay for a more structural and complete discussion.
- **Appendix D**: We introduce the setting of **large-scale continual pretraining experiment** (**3.2B model**, ~730B tokens in base phase, **over 150B tokens** in continual phase), and show details of our implementation and results. (`vPHj` `QrZN`)
- **Appendix E**: We strengthen the discussion of our work against various previous works to better position our work relative to the precedents. (`qZNE`)
- We also fine-tuned the figures and captions to make them easier to understand. In particular, we adjusted `Figure1(d)` and `Figure3[left]` to better align with the section contents.

We hope these revisions address your concerns and will facilitate your reference in the following discussion.

---

### Author Response · Authors · 2025-11-26
**General Response [1/2]**

We sincerely thank all the reviewers for taking the time to review our paper and for engaging with the content of our work. We appreciate your interest in the curriculum model average (CMA/CDMA), and your insights and feedback have been crucial in helping us improve our work. We have slightly reorganized the paper and added additional experiments based on the reviewers' suggestions, as summarized in the `Summary of Revisions`.

We note that reviewers `vPHj` and `QrZN` raised questions about verifying our strategy on larger model sizes or longer training runs. We conducted additional experiments in the following setting to provide a more comprehensive and convincing verification of CMA/CDMA and address the reviewers' concerns.

---

**Q: Verification of CMA/CDMA at a larger scale of model/data size. (`vPHj`;`QrZN`)**


**A: We applied our strategy to a multi-domain continual pretraining regime, training a 3.2B model over 150B tokens**, after resuming from a checkpoint trained over ~729B tokens in the base phase. **Our results achieve a 1.09% improvement using the same dataset as the baseline at this scale.** The baseline adopts uniform data ordering and standard LR decay, while our approach uses a multi-domain curriculum and model average, with a moderate LR decay (CDMA). The training dataset comprises a data mixture, and we believe this setting is close to a practical pretraining setup. In this experiment, we use the architecture of Qwen2.5-3B, with a total of 3.2B parameters (without tied word embedding). The details of our experiment setup can refer to Appendix D.2 in our paper revision.

**Base Phase:** In the base phase, we train on approximately 729B tokens, using a mixed dataset that includes deduplicated DCLM-Baseline (DCLM) [1], Fineweb-Edu-Chinese-V2.1 (Fineweb-C) [2], Starcoder dataset [3], and MegaMath [4], following uniform sampling. In the base phase, we use a constant LR at $1 \times 10^{-3}$ after warmup, with a batch size of 2048.

**Uniform Mixture Details**

| Dataset   | Token Count (B) | Ratio |
|-----------|------------------|-------|
| DCLM      | 608.5B           | 83.51% |
| Fineweb-C | 91.8B            | 12.60% |
| Starcoder | 19.0B            | 2.61% |
| MegaMath  | 9.3B             | 1.28% |
| **Total** | **728.7B**       | 100% |

---

### Author Response · Authors · 2025-11-26
**General Response [2/2]**

**Continual Phase:** In the continual phase, we use about 156.3B tokens and select top samples from each domain: by quality scores (DCLM, Fineweb-C), by max star counts (StarCoder), and by duplicate count (MegaMath).

The baseline method uses uniform data ordering. We use a standard WSD schedule, with a continual stable phase for 72.2B tokens and the decay phase consisting of the final 83.9B tokens. In this way, we align our decay ratio of the WSD schedule with previous work [5]. The baseline linearly decays to $1 \times 10^{-5}$ in the decay phase.

For our strategy, we adopt a multi-domain data curriculum, which keeps the data curriculum (ascending quality score) within each domain, while maintaining the domain ratios stable throughout training. The implementation details can refer to Appendix D.1 in the paper revision. We adopt a similar WSD schedule as the baseline, except that we keep the LR after it decays to $3.67\times 10^{-4}$, using a more moderate LR decay. Finally, we average the last six checkpoints (checkpoint interval is about 1.67B tokens), and use $\alpha = 0.2$ with EMA to obtain the final checkpoint. Due to time and compute limits, we have not tuned hyperparameters extensively.

**Continual Phase Mixture Details (Top-k Selection)**

| Dataset   | Token Count (B) | Count Ratio | Top-k Ratio | Score Metric      |
|-----------|------------------|-------------|------------------------|------------------|
| DCLM      | 83.9B            | 53.77%      | 0.2               | fasttext score    |
| Fineweb-C | 23.9B            | 15.34%      | 0.2               | fineweb score     |
| Starcoder | 38.9B            | 24.95%      | 0.2               | max stars count   |
| LLM360    | 9.3B             | 5.94%       | 0.4               | duplicate count   |
| **Total** | **156.1B**       | 100%        | -                 | -              |

**Benchmark Evaluation Results**

The resulting benchmark table is as follows:

| Method   | GSM8K                     | sanitized MBPP          | CEval                     | MMLU                      | ARC-Challenge             | ARC-Easy                | CSQA        | BoolQ                     | Average              |
| -------- | ------------------------- | ----------------------- | ------------------------- | ------------------------- | ------------------------- | ----------------------- | ------------------------- | ------------------------- | ------------------------- |
| Baseline | 24.64                     | **38.91**               | 40.83                     | 48.53                     | 51.86                     | **78.13**               | 67.32                     | 71.38                     | 52.70                     |
| Ours     | **24.72**(+0.08) | 38.13(-0.78) | **42.12**(+1.29) | **48.97**(+0.44) | **55.25**(+3.39) | 77.95(-0.18) | **68.39**(+1.07) | **74.77**(+3.39) | **53.79**(+1.09) |

Our CDMA strategy with a multi-domain curriculum outperforms the baseline in large-scale continual pretraining on average. We observe significant improvements across multiple capability dimensions: general knowledge (MMLU: `+0.44`), commonsense reasoning (CSQA: `+1.07`), reading comprehension (BoolQ: `+3.39`), and complex reasoning (ARC-Challenge: `+3.39`). Notably, our approach also demonstrates strong performance on Chinese language understanding (CEval: `+1.29`), indicating generalization of our approach across different linguistic domains with appropriate quality metrics. **The overall core average improvement of 1.09% represents a meaningful advancement at this scale.**

However, we note limited gains in mathematical reasoning (GSM8K: `+0.08`) and a slight regression in coding capability (MBPP: `-0.78`). The relatively lower mixing ratios can limit the benefits in these domains. Besides, we hypothesize that these domains may benefit from more sophisticated quality metrics than the preliminary measures (e.g., GitHub star counts) employed in this work. This direction presents an opportunity for future research.

[1] [DataComp-LM: In search of the next generation of training sets for language models](https://arxiv.org/abs/2406.11794)

[2] [OpenCSG Chinese Corpus: A Series of High-quality Chinese Datasets for LLM Training](https://arxiv.org/abs/2501.08197)

[3] [StarCoder: may the source be with you!](https://arxiv.org/abs/2305.06161)

[4] [MegaMath: Pushing the Limits of Open Math Corpora](https://arxiv.org/abs/2504.02807)

[5] [WSM: Decay-Free Learning Rate Schedule via Checkpoint Merging for LLM Pre-training](https://arxiv.org/abs/2507.17634)

---

### Meta-Review · Area_Chair_21eb · 2026-01-06

**Summary:**

Reviewers generally found the paper technically sound, clearly written, and empirically convincing, with a strong and intuitive diagnosis of the incompatibility between curriculum learning and standard learning-rate decay schedules in LLM pretraining. The primary concern raised across reviews was the limited experimental scale in the original submission, which focused mainly on a 1.5B-parameter model trained on 30B tokens, leaving some uncertainty about generalization to larger or more realistic pretraining settings. In the rebuttal, the authors addressed this concern by adding large-scale continual pretraining experiments, scaling to a 3.2B-parameter model trained on ~150B additional tokens, starting from a checkpoint pretrained on ~729B tokens. These new results show consistent improvements across multiple benchmarks and domains, substantially strengthening the empirical support for the paper’s core claims and supporting acceptance.

**Reviewer Concerns:**

**Concerns largely addressed in the rebuttal**

- **Limited experimental scale** (vPHj – Weakness; QrZN – scale/generalization concern):
  The original submission focused mainly on a 1.5B-parameter model trained on 30B tokens. In the rebuttal, the authors added large-scale *continual pretraining* experiments, scaling to a 3.2B-parameter model trained on ~150B additional tokens, starting from a checkpoint pretrained on ~729B tokens. These results demonstrate consistent improvements across multiple benchmarks and domains, substantially strengthening confidence in the generality of the main conclusions.

- **Hyperparameter sensitivity of model averaging** (vPHj – Weakness):
  The rebuttal includes extensive ablation studies on checkpoint interval, number of checkpoints, and EMA decay factor. The results show that CMA/CDMA performance is reasonably robust across these choices and provide clear justification for the selected defaults.

- **Missing high-quality data repetition baseline** (qZNE – Weakness):
  The authors added Filter&Repeat baselines under matched compute budgets. While aggressive filtering and repetition can improve some individual benchmarks, these baselines underperform CMA/CDMA on average, supporting the claim that curriculum design provides benefits beyond simple data repetition.

- **Limited analysis of descending curricula** (qZNE – Weakness):
  The rebuttal adds targeted experiments and analysis disentangling learning-rate matching and forgetting effects, clarifying why Descend+Decay underperforms in practice and strengthening the paper’s empirical narrative.

**Remaining limitations**

- **Very large model scales (e.g., ≥7B parameters)** (vPHj – Weakness/Question):
  While the added 3.2B continual pretraining experiments substantially improve empirical validation, the paper does not include results at ≥7B scale. Given the pretraining focus and academic setting, this limitation is reasonable but remains unaddressed.

- **Comparison with learned or adaptive curricula** (QrZN – Weakness):
  The paper does not evaluate learned or adaptive curriculum methods. The authors justify this as out of scope for their diagnostic focus, but this remains an open direction for future work.

**Reviewer Scores:**

- **Reviewer vPHj**:
  The main concerns were the limited experimental scale and lack of hyperparameter analysis. Both were directly addressed in the rebuttal through large-scale continual pretraining experiments (3.2B model, ~150B additional tokens) and extensive ablations on model averaging hyperparameters. As a result, vPHj would likely maintain their original score or increase it slightly (e.g., from 6 to 7).

- **Reviewer qZNE**:
  Key concerns regarding missing baselines (high-quality data repetition), limited analysis of descending curricula, and positioning against prior work were all addressed with new experiments, deeper analysis, and expanded discussion. Additionally, issues related to questionable references in the review were clarified by the authors. Overall, qZNE would likely increase their score relative to the original assessment.

- **Reviewer QrZN**:
  The reviewer’s main concern was validation at larger scale and generalization across domains and quality metrics. The rebuttal added a substantially larger-scale continual pretraining experiment and demonstrated gains on both English and Chinese benchmarks, directly addressing these points. QrZN would likely keep their original score or increase it (e.g., to around 7).

Overall, based on explicit post-rebuttal evidence and reviewer concerns being largely addressed, a reasonable post-discussion interpretation of the reviews is approximately **7, 7, and 7**, corresponding to a **clear accept profile**.

---

### Decision · Program_Chairs · 2026-01-26

Accept (Oral)